



# Paleogeographic numerical modelling of marginal seas for the Holocene – an exemplary study of the Baltic Sea

Jakub Miluch[1,2], Wenyan Zhang[1], Jan Harff[2], Andreas Groh[3], Peter Arlinghaus[1], Celine Denker[1]

[1]Institute of Coastal Systems – Analysis and Modeling, Helmholtz-Zentrum Hereon, Geesthacht, 21502, Germany
[2]Institute of Marine and Environmental Sciences, University of Szczecin, Szczecin, 70-453, Poland
[3]Institute of Planetary Geodesy, Technical University Dresden, 01062, Germany

*Correspondence to*: Wenyan Zhang (wenyan.zhang@hereon.de); Jakub Miluch (jakub.miluch@usz.edu.pl)

**Abstract.** Sustainable management of marginal seas is based on a thorough understanding of their evolutionary trends in the past. Paleogeographic evolution of marginal seas is controlled by not only global and regional driving forces (eustatic sea level change and isostatic/tectonic movements) but also sediment erosion, transport, and deposition at smaller scales. Consistent paleogeographic reconstructions at a marginal sea scale considering the global, regional and local processes is yet to be derived, and this study presents an effort towards this goal. We present a high-resolution (0.01°×0.01°)

paleogeographic reconstruction of the entire Baltic Sea and its coast for the Holocene period by combining eustatic sea-level change, glacio-isostatic movement, and sediment deposition. Our results are validated by comparison with field-based reconstructions of RSL and successfully reproduce the connection/disconnection between the Baltic Sea and the North Sea during the transitions between lake and sea phases. A consistent map of Holocene sediment thickness in the Baltic Sea has been generated, which shows that relatively thick Holocene sediment deposits (up to 36 m) are located in the southern and

central parts of the Baltic Sea, corresponding to depressions of sub-basins including the Arkona Basin, the Bornholm Basin as well as the Eastern and Western Gotland Basins. In addition, some shallower coastal areas in the southern Baltic Sea also host locally confined deposits with thickness larger than 20 m and are mostly associated with alongshore sediment transport and formation of barrier islands. In contrast to the southern Baltic Sea, the Holocene sediment thickness in the northern Baltic Sea is relatively thin and mostly less than 6 m. Morphological evolution of the Baltic Sea and its coastline is featured

by two distinct patterns. In the north-eastern part, change of the coastline and offshore morphology is dominated by regression caused by post-glacial rebound that outpaces the eustatic sea level rise, and the influence of sediment transport is very minor, whereas a transgression together with active sediment erosion, transport and deposition have constantly shaped the coastline and the offshore morphology in the south-eastern part, leading to formation of a wide variety of landscapes and seascapes such as barrier islands, spits and lagoons.



## 1. Introduction

The majority of the Earth's coasts and shelf seas are experiencing dramatic morphological changes as a result of joint effects of natural processes and anthropogenic activities (Mentaschi et al., 2018). Coasts of marginal seas composed of erodible soft material (sands, mud and moraine) are most variable in this context (Harff et al. 2017; Luijendijk et al., 2018; Hulskamp et al., 2023). At short time scales, their morphology is constantly reshaped by atmospheric and oceanic forcing

such as winds, tides and waves, and human interventions (Zhang et al., 2011a; Mentaschi et al., 2018; Weisse et al., 2021). At longer time scales, climate change-induced oscillations of sea level, ice-cover/retreat, isostatic / tectonic movements and variations in sediment supply exert a major control on morphological development of marginal seas (Zhang and Arlinghaus, 2022). In the forthcoming centuries, relative sea-level rise will increasingly influence coastal morphological change and challenge the defence of coastlines. Sustainable management of marginal seas therefore requires a thorough understanding of

the past and future morphological trends of evolution (Hulskamp et al., 2023). Management strategies need to consider the "geo-environmental" change in the past and future to separate natural and anthropogenic driving forces (Neumann et al., 2015). Learning from paleo-geomorphological history, particularly the post glacial period, will help to understand the coastal change in future (Harff et al., 2017).

Paleogeographic evolution of marginal seas is strongly associated with global and regional driving forces. Global

climatically controlled eustatic changes (Gale et al., 2002; Berra et al., 2010) interplay with regional settings such as tectonics (Watts 1982; Vött 2007), isostasy (Peltier 1999, 2007; Lambeck 2010, Spada et al., 2012) or even local factors like sediment dynamics (Einsele 1996). Combination of these overlapping forces influences the relative sea level and may lead to significantly different coastal behaviours in various sections of marginal seas due to spatial heterogeneity of described forces intensity (Rosentau et al., 2021).

Most paleogeographic reconstructions of marginal seas are based on a reversal of relative sea level composed of eustatic sea level change plus tectonic and glacio-isostatic crustal effects (Uehara et al., 2006; Harff et al., 2007; Yao et al., 2009; Sturt et al., 2013). A few studies have additionally incorporated sediment relocations based on empirical interpolation functions and information from dated sediment cores, but are limited to local areas (Zhang et al., 2014; Xiong et al., 2020; Karle et al., 2021). This study presents a high-resolution paleogeographic reconstruction of the entire Baltic Sea and its

coasts for the Holocene period by taking into account not only eustatic sea-level change and glacio-isostatic movement but also sediment dynamics. As such, this work represents a further step for comprehensive paleogeographic reconstruction of marginal seas resolving spatial heterogeneity of driving forces across multi-scales. Our motivation is twofold: first to depict the morphological evolution of a complex marginal sea system in response to the impact of climate change and oceanic sedimentation; second, to provide a sediment budget analysis of the marginal sea for the Holocene period and compare with

present-day sediment fluxes from land to the sea in order to disentangle natural and anthropogenic impacts.



## 2. Geological setting

The Baltic Sea is a semi-enclosed intra-continental marginal sea, connected with the North Sea through the Danish Straits and the Swedish Sound (Rosentau et al., 2017). In terms of regional tectonics, the Baltic Sea Basin bridges between the Eastern European Platform consisting of the Fennoscandian (Baltic) Shield in the NE and the Russian Plate in the SE and the Western European Platform in the SW. Eastern and Western European Platform are separated by the deep NW-SE striking tectonic fault system of the Tornquist-Teisseyre Zone (TTZ) and its northwestern prolongation, the Sorgenfrei-Tornquist Zone (STZ) (Uścinowicz 2014). Northeast of this zone, Precambrian crystalline rocks of the Baltic Shield and undeformed Phanerozoic sediments of the Russian Plate on Precambrian basement form the coastal frame of the Baltic Sea. West of the TTZ, the Central Caledonides and Variscides together form the deep sedimentary basin of the Central European Depression filled mainly with Mesozoic deposits on a basement at depth of 10-15 km (Uścinowicz 2014). The lowlands are mainly covered by Pleistocene sediments consisting of glacial, glacio-fluviatile and lacustrine deposits. The Holocene is represented by lacustrine and brackish marine deposits (Rosentau et al., 2017). Glaciers have shaped the surface of the mainland surrounding the Baltic Sea, as well as the Baltic Sea Basin itself (with an average water depth of 55 m) where they formed a series of sub-basins separated by shallower sills (Hall and van Böckel 2020). Danish Straits and the Sound connect the Baltic Sea permanently with the North Sea. Humid climate and a positive water balance serve for an estuarine circulation and a stratified water body with remarkable vertical and horizontal differences in salinity, density, and temperature of the water body (Matthäus and Franck, 1992; Wulff et al., 1990).

Advance and retreat of inland ice during the Quaternary glacial cycles have caused a change of isostatic loading and unloading of the Baltic Basin's crust leading to a cyclicity in vertical crustal movement correlated to the climate cycles. The relationship between vertical crustal movement and sea-level change determines the hydrographic communication of the open sea and the inner Baltic Basin. If the land uplift exceeds the rise in sea level, this means that the basin is closed off, while land subsidence exceeding sea-level drop leads to the connection of the open sea with the basin. The gates between these basins thus take on an isostatic/hydrographically controlled "gate function". Correspondingly, the paleogeographic history of the Baltic Basin during the Quaternary was ruled mainly by the glacial cycles following Milankovitch cyclicity. However, because of the erosional effects of the several times advancing inland ice sediments reflecting by proxy-data the geological history remained just from the post-glacial period. Andrén et al. (2011) have depicted this postglacial history based on the interpretation of proxy data, such as basin sediments and paleo-coastlines by a set on paleogeographic maps (Fig. 1) which are used here for plausibility checks of the results achieved by numerical modeling.

The evolution of the Baltic Sea since Last Glacial Maximum (Fig.1) is characterized by shift events between fresh water and marine sedimentary environments driven by an interplay of eustatic and glacio-isostatic forces (Andren et al., 2011). Starting from 16 kyr BP the Baltic Ice Lake (BIL) was formed in front of the retreating ice sheet filled with glacial melt water. Initially the water level in the lake was similar to global level and the melt water discharged to the Baltic Basin was flowing through the Öresund into the North Sea. The drainage has however ceased due to glacio-isostatic uplift of the



Öresund area around 14 kyr BP (Björck 2008). Another drainage event occurred 13 kyr BP via the Central Swedish

Lowlands as a result of rising water level caused by meltwater discharge to the BIL, although ca. 12.8 kyr BP a short cooling

phase of the Younger Dryas caused glacial readvance and interrupted the water exchange between the Baltic Basin and the

Paleo-North Sea (Björck, 1995). As the isostatic uplift rate of the Baltic Shield was generally exceeding the sea-level rise,

the Baltic Basin remained disconnected from the Paleo-North Seas so that the Ice Lake stage lasted until the beginning of

Holocene 11.7 kyr BP (Jakobsson et al., 2007). Inland ice retreat and global sea-level rise re-opened the gate connecting

Baltic and Paleo-North Sea Basin and via Central Swedish Lowlands, initiating the brackish-marine Yoldia Sea stage

(Heinsalu and Veski 2007). At 10.7 kyr BP, isostatic uplift, again surpassing the eustatic rise, re-closed the gate of the

Swedish Depression, and the Baltic Basin turned again to a freshwater (lake) environment (Sohlenius et al., 2001). The so-

called Ancylus Lake stage was characterized by continuously rising water level because of meltwater discharge from inland

ice remains until it reached its maximum at 10.5 kyr BP, to decrease afterwards because of drainage processes to the Paleo-

North Sea (Lemke et al., 2001; Rosentau et al., 2014). The global Holocene sea-level rise in connection with the collapsing

lithospheric bulge surrounding the Baltic Shield opened the pathway of the Littorina Transgression at 8.0 kyr BP that

connected the Baltic Basin permanently to the North Seas.

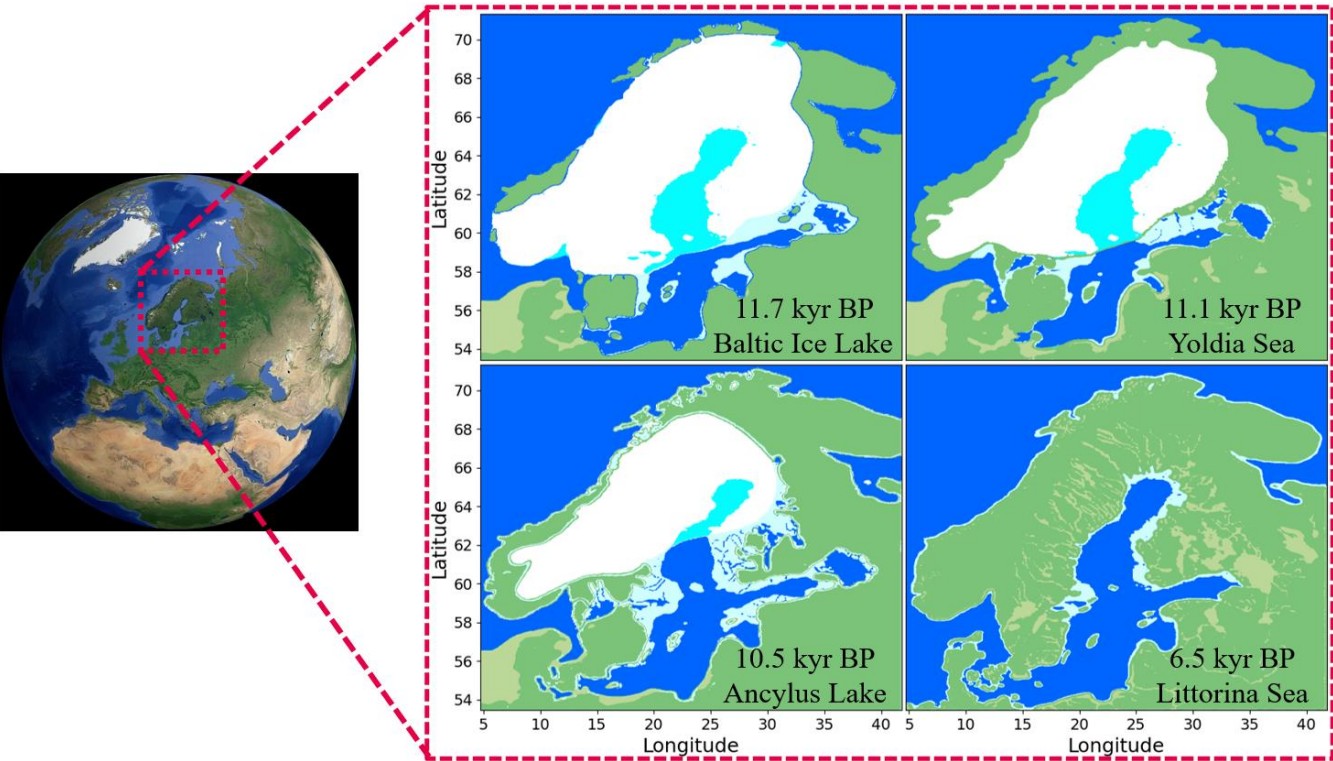

**Figure 1:** Location of the Baltic Sea (© Google Earth Pro) and paleogeographic maps representing different Holocene stages

of the Baltic Sea (modified after Andrén et al. (2011): namely the Baltic Ice Lake (just prior to the final drainage) at 11.7 kyr



BP, the Yoldia Sea (end of the brackish phase) at 11.1 kyr BP, the Ancylus Lake (transgression maximum) at 10.5 kyr BP, and the Littorina Sea (most saline phase) at 6.5 kyr BP. Ice cover is shown by the mask in white.

## 3. Data and methods

The main target of this study was the generation of a set of paleo-Digital Elevation Models (paleo-DEMs) corresponding to high-resolution, in both time and space, reconstructions of stages of Baltic Sea evolution. Such reconstructions play a critical role for the purpose of basin analysis (Allen and Allen, 2008) leading to visualisation of the past state of the basin (Xiong et al., 2020), morphogenetic interpretation (Miluch et al., 2021, Miluch et al., 2022) as well as hydro-morphodynamic modeling of circulation and sediment transport systems (Zhang et al., 2020). For the purpose of

investigation of the Baltic Sea basin evolution, the key factors are: eustatic sea level variations, isostatic vertical crust movements and sediment budget (Harff et al., 2007; Yao et al., 2009; Eq 1). As the study involves reconstruction of the Baltic Sea basin for a relatively short geological time span - from the beginning of Holocene (11.7 kyr BP) till present day, other variables such as sediment compaction which play a secondary role in such time scale are neglected here. The conceptual equations (Harff et al., 2017) below were applied to generate the paleo-DEMs:

$$DEM_t = DEM_0 - \Delta RSL + \Delta SED, \tag{1}$$

where $DEM_t$ is the Paleo-digital elevation model at time $t$, $DEM_0$ is the present day digital elevation model, $\Delta RSL$ is the relative sea-level change, and $\Delta SED$ is the change of sediment thickness by accumulation or erosion.

The relative sea-level change is described in the following formulation following Harff et al. (2017) by neglecting minor effects by gravitational forcing:

$$\Delta RSL = \Delta EC + \Delta GIA + \Delta SED, \tag{2}$$

where $\Delta EC$ is the global eustatic sea-level change referred to current sea level $EC_0$, and $\Delta GIA$ refers to the Glacio-isostatic adjustment.

Expressions used in equations (1) and (2) stand for variables with values assigned to nodes of one and the same discrete georeferenced grid. After successful application of the conceptual equations in the South China Sea by Yao et al.

(2009) and Xiong et al. (2020), we describe the first time its application to the Baltic Basin considering not only GIA and sediment accumulation but also differences in eustaic sea level and the water level in a regional basin following temporarily a separate regional hydrographic regime when it is disconnected from the open sea.

During the brackish-marine stages of the Baltic Sea, a 1:1 sea-level transfer function from the North Atlantic to the Baltic Sea basin is assumed. EC thus reflects the global (eustatic) sea level in the entire study area (see Section 2, Geological-tectonic evolution of the Baltic Basin). When the Baltic Sea basin is decoupled from the World Ocean, the

variable EC reflects the level of the relevant freshwater lake that fills the Baltic Sea basin and is fed exclusively by precipitation and inland ice's meltwater. In this case, the areas of the "outer coast" of Fennoscandia influenced by the North



Sea, Norwegian Sea and Barents Sea and must be modeled separately from the "inner" Baltic Sea (lake) basin. The Baltic
Sea basin to be modeled is during the lake phases separated by watersheds on the surrounding mainland.

The models were numerically handled and resulting maps were plotted using Golden Software Surfer 18 program.
The software allows to visualize surface relief with pre-defined spatial resolution (Bola and Kayode, 2014; Libina and
Nikiforov, 2020), perform grid-on-grid mathematical operations (Liu et al., 2020) as well as interpolate the data (Gonet and
Gonet, 2017; Razas et al., 2023; Yilmaz 2007), and therefore acts as a convenient, widely-used tool in basin analysis
(Covington and Kenelly, 2018; Grunt and Geiger, 2011). The general workflow for data collection, synthesis and
interpretation is depicted in Fig. 2.

**Figure 2:** General workflow chart for data collection, synthesis and interpretation.



### 3.1. Digital Elevation Model (DEM$_0$)

The initial grid acting as a base for paleogeographic reconstruction was obtained from the Global Bathymetric Chart of the Ocean (Gebco) (Sandwell et al., 2002; Becker et al., 2009). From the present-day "amphibious" Digital Elevation Model (DEM$_0$), consisting of both bathymetric and onshore data (GEBCO, 2023), Baltic Sea basin region was extracted, extending from 9.5 to 31ºE and from 52 to 66.5ºN. The initial spatial resolution of the grid was diminished and set to 0.01 degree. Maintaining original resolution was unnecessary as other components of the conceptual paleogeographic modeling equation (eustatic and isostatic gradients) were characterized by significantly lower grid resolution. Such transformation had

negligible influence on the quality of the generated maps, in parallel allowing to boost the map generation speed and save storage space.

### 3.2. Eustatic data (EC)

For the generation of regional sea level curve, a dataset by Waelbroeck et al. (2002) was used as a base. However, adjustment to the regional paleogeographic setting is needed. The Baltic water level curve followed the Waelboeck et al.

(2002) data for the brackish-marine stages, during which the Baltic water was connected with the North Sea – 11.7 kyr BP to 11 kyr BP (Yoldia Sea) and 9.5 kyr BP until present (Littorina Sea), respectively (Andren et al., 2011). Knowing that during the Ancylus Lake stage (11 to 9.5 kyr BP) water (lake) level in the Baltic water was generally higher than in the global ocean, a local sea level dataset of the southern Baltic region by Uścinowicz (2006) was applied to adjust the curve correspondingly. Even though the local relative sea level is strongly influenced by the Glacio-Isostatic Adjustment (Andren

et al., 2002, 2011; Groh et al., 2017; Rosentau et al.,2012, 2021; Harff et al., 2017), the southern Baltic region is located near the isostaticallly neutral hinge-line between uplift in the north and subsidence in the south of it (Statteger and Leszczyńska, 2023). Therefore glacio-isostatic adjustment rate played a minor role in estimation of the relative lake level curve in the southern Baltic region. Such feature allowed to incorporate sea level values of the Ancylus Lake from Uścinowicz (2006) as "eustatic component" into the RSL conceptual equation (eq. (2)). Moreover, lake phase of the Baltic water required a

subdivision into two independent water level systems: Ancylus Lake and the North Sea separated by closed Danish Straits. The border between these two water bodies was set based on the paleo-catchment areas obtained using "Terrain Aspect" function in Golden Software Surfer (Golden Software Surfer) on previously generated paleo-DEMs. In the lake phase, the eustatic water level in the Baltic basin followed the Uścinowicz (2006) curve, whereas the North Sea followed Waelbroeck et al. (2002). Fig. 3 depicts the merged sea-/water- level curve for the Baltic Basin during the Holocene.






**Figure 3:** Holocene water level curve for the Baltic generated by combination of Waelbroeck et al. (2002) and Uścinowicz (2006).

## 3.3. Vertical crustal movements - Glacio-Isostatic Adjustment (GIA) including paleo-ice-thickness model

The solid Earth's response to the changing surface loads of the vanishing Pleistocene ice sheets is known as glacial isostatic adjustment (GIA). This visco-elastic response comprises an instantaneous elastic and a delayed viscous component. GIA manifests in terms of deformations of the Earth's crust, changes in relative sea-level, i.e. sea level with respect to the Earth's deformable crust, and changes in gravitational potential (e.g. Peltier 1998). The later originates from the redistribution surface masses, i.e. the melting of ice masses and the fresh water consequently added to the ocean, as well as

from material in the Earth's mantle relocated as a reaction to the changing surface loads.



The decreased stress on the Earth's crust induced by the melting of the Laurentian Ice Sheet after last glacial maximum (LGM) about 21 kyr BP, caused a crustal deformation which is still ongoing at present. The crust is uplifting in regions formerly covered by the ice load. Crustal subsidence can be observed in regions surrounding the former ice cover known as the peripheral bulge. This subsidence originates from the relocation of mantle material back to its original location

in the Earth's interior underneath the region formerly covered by the ice load. The crustal uplift is partly compensated by the additional water load stemming from the fresh-water influx. However, the melt water is unevenly distributed over the ocean according to the changing gravitational potential caused by the varying distribution of ice masses as well as by the water masses themselves. Thus, close to the melting ice sheets, a drop in relative sea-level can be observed due to both the decreasing gravitational attraction of the ice and the uplifting crust.

The complex interaction between changing ice and water loads, their gravitational potential as well as the induced crustal deformations is described in a gravitational self-consistent way by the sea-level equation (e.g., Farrell and Clark, 1976; Peltier, 1998). To solve the sea-level equation in order to model the GIA-induced vertical crustal deformation we make use the freely available software package SELEN (Spada and Stocchi, 2007). This software makes use of the ICE-5G ice load history (Peltier, 2004) to describe the spatio-temporal evolution of the ice sheets from LGM until present day at a

temporal resolution of 1 kyr. Figure 4 depicts the ice thickness at LGM (21 kyr BP) according to the ICE-5G load history as implemented by the SELEN software package. Cumulated crustal deformations are modelled starting from an equilibrium state at LGM. Details of the applied model set-up are provided in Groh and Harff (2023).






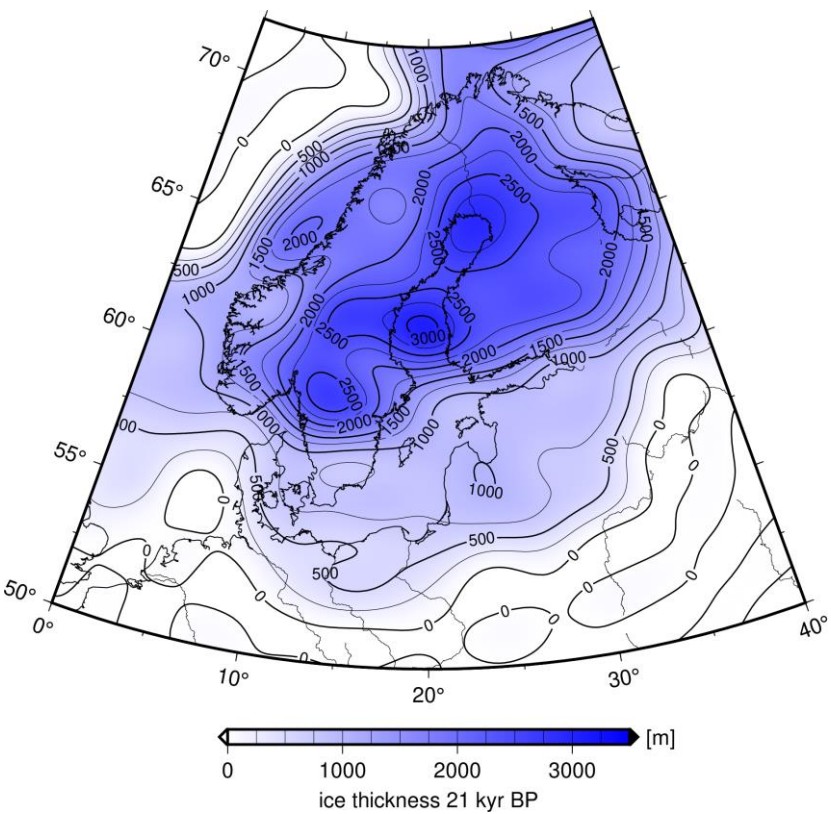

**Figure 4.** Present-day coastline and ice thickness at the Last Glacial Maximum (21 kyr BP) according to the ICE-5G ice load history (Peltier, 2004) as implemented by the software package of Spada and Stocchi (2007) after synthesising its spherical harmonic representation to space domain.

## 3.4. Sediment thickness (SED)

The Baltic Sea Basin is administrated by nine countries: Denmark, Germany, Poland, Sweden, Lithuania, Latvia, Estonia, Finland and Russia. Due to a fact that no joint Holocene sediment thickness database is available, it is necessary to identify and merge the existing local-scale datasets from various national or publicly available sources. In order to generate a regional sediment thickness model, eight local datasets were synthesized. These sub-datasets were derived by six methods of data acquisition: map digitalization, recalculation of Holocene bottom depth maps, interpretation of 2-D seismo-acoustic

profiles and data interpolation, interpolation of point data from sediment cores, extrapolation using co-kriging, and extrapolation using convolutional neural network machine learning method. The areas with corresponding data acquisition methods are shown in Fig.5.



**Figure 5:** Map marking different types of data sources with references for acquisition of sediment thickness sub-datasets
collected by: digitalization of thickness maps (red), recalculation of thickness from seismic reflector depth maps (green),
digitalization of profiles and data interpolation (dark blue), interpolation of point data (black); as well as areas with data derived
from extrapolation using co-kriging (yellow) and machine learning (bright blue).

### 3.4.1. Southern Baltic

The southern Baltic Sea area is characterized by significant Holocene sediment thickness and satisfactory data
quality. The sediment thickness model was generated as a result of merging three local-scale grids. German and Danish parts
were covered by data derived from Lemke (1998). Maps representing the depth of top of glacial till were assumed to
correspond to the Pleistocene basement. This gridded dataset was thus "subtracted" from the present-day bathymetry
(GEBCO, 2023) in order to estimate the sediment thickness. The thickness model for the Polish waters was provided by





Uścinowicz (1998), whereas data for the Russian (Kaliningrad area) and Lithuanian territories was retrieved from
Emelyanov (2002). Due to slight differences in thickness values on the border between these two datasets, the values at the
grid junction were averaged.

### 3.4.2. Central Baltic

The central Baltic (Bornholm Basin, East Gotland Basin, and its surroundings) Holocene thickness grids were
retrieved from 2-D seismo-acoustic profiles collected by Leibniz-Institute for Baltic Sea Research Warnemünde, Germany
(courtesy of Dr. Peter Feldens). The Holocene basement corresponds mostly to the top of glacial till (Late Pleistocene age,
non-stratified seismic facies) as well as to bottom of glacial varved sediments (Baltic Ice Lake – early Holocene age,
stratified seismic facies). The seismic reflector corresponding to pre-Holocene basement was exported as depth data points
(= shot points), recalculated from two-way travel time to metric units assuming sound velocity in sediment to 1600 m/s, and
further interpolated using ordinary kriging (Wackernagel and Wackernagel, 2003). Obtained basement subsurface depth grid
data were subtracted from GEBCO bathymetry (GEBCO, 2023) in order to generate the Holocene sediment thickness model.

### 3.4.3. Eastern Baltic

Holocene sediment thickness data grid of the Gulf of Riga as well as of the Gulf of Finland was generated based on
profiles provided by Gudelis and Emelyanov (1976). The reflector corresponding to the top of glacial till was digitized and
exported as data points, interpolated using ordinary kriging, and subtracted from the GEBCO bathymetry (GEBCO, 2023).
Thickness data grid covering the Russian part of the Gulf of Finland was derived from the digitized sediment thickness map
by Ryabchuk et al. (2020).

### 3.4.4. Northern Baltic

Holocene thickness data in the northern Baltic is generally scarce because the area is under sampled. A dataset by
Winterhalter (1972), consisting of several polygons, located in the Bothnia Sea and representing mean thickness served as a
base for interpolation. The central point of each polygon was exported as a data point and then modelled using ordinary
kriging. To solve the problem caused by the severely under sampled northernmost Bothnian Bay, data points of thickness
from Winterhalter (1972) were used for extrapolation by co-kriging (Goovaerts, 1998; Myers 1982, 1984). The method is
commonly used in environmental science, geology, and other fields where correlated measurements of multivariate variables
are available (Belkhiri et al. 2020; Leenaers et al. 2020; Konomi et al., 2023). In the northern Baltic case, a two-dimensional
variable consisting of a GIA-corrected paleo-bathymetry and thickness of Holocene sediments was used. Parameters of the





corresponding semi-co-veriogram were determined using those gridnodes where data of both variables were available. These parameters allowed a co-kriging estimation of thickness data for undersampled areas.

### 3.4.5. Filling of data gaps

Machine learning was applied to fill the remaining gaps in the Holocene sediment thickness data. The areas with data gaps mainly include the central western part of the Baltic Sea (see Fig. 5). A convolutional neural network (CNN), namely a *U-Net* (Ronneberger et al., 2015) was build using *PyTorch* (Paszke et al., 2019). *U-Net*s have been proven to be a robust and versatile tool for image and data analysis (Zhang et al., 2018; Liu et al., 2020) and are superior to pixel-based methods such as random forest (Boston et al., 2022). In this study four variables, namely the paleo-land-surface morphology, the median grain size of surface sediments and longitude and latitude values were used to predict sediment thickness. The reason for adding the latter two variables is their relationship with the GIA (Fig.4). The available data were randomly cut into 420 squares of 32×32 pixels size, excluding the Gulf of Finland and the Gulf of Bothnia. The reason for excluding the two regions is because they differ substantially in geological, tectonic, and depositional environment from the central, south, and southwest parts of the Baltic Sea (Harff et al., 2017). This helps to reduce the error in the predictions. In the 420 sub-datasets, 80% were used for model training and 20% for assessment of the model prediction. The input of the *U-Net* has the shape (32, 32, 4) and the output shape is (32, 32). The first layer consists of a double convolutional block performing 3×3 convolution with 64 output channels, padding, batch normalization and ReLU activation. The training was performed with 100 epochs and the mean squared error (MSE) was calculated as the loss function (*torch.nn.MSEloss*). Re-running the model with different random initializations and dropout yields different model results with the same general pattern but some local differences in sediment thickness. The result with the smallest value of MSE (6.1 m²) was chosen (Fig.6). This corresponds to an average deviation from the validation to the measured data of 5.8%.

### 3.4.6. Merging thickness grids

Each local Holocene thickness grid was unified to identical resolution of 0.01×0.01 degree and merged into one regional grid. In case of small spatial gaps between the sub-datasets the thickness was linearly interpolated. In case of dataset not covering the coastline, the solution proposed by Miluch et al. (2021) was used by setting equidistant pinpoints characterized by 0-thickness value along the coastlines and included into the dataset. Such solution allowed to linearly extrapolate the thickness data between the grid and the coastline in order to completely close the remaining data gaps.





### 3.5. Sediment budget analysis

Generation of thickness map allowed to perform sediment budget analysis. First, the grid was transformed from geographic coordinate system to UTM projection in order to conduct volume calculation scaled in metric units. Applied volume calculation algorithms include the trapezoidal rule, the Simpson's rule and the Simpson's 3/8 rule (Atkinson, 1989). Result from each algorithm was compared to assess the uncertainty. Sediment mass modeling required taking into account sediment bulk density, being correlated with sediment particle density and porosity. Surface sediment porosity was calculated based on the present-day sediment grain size map of the Baltic Sea derived from the project DYNAS (Dynamics of natural and anthropogenic sedimentation) (Bobert et al., 2009; Harff et al., 2011) and application of the empirical formula from Endler et al. (2015) to link porosity to the median grain size. Knowing that compaction rates of sandy and silty sediments is neglectable at a thickness scale of meter (Schmedemann et al., 2008), constant vertical porosity depending on the local grain size was assumed for the Holocene deposit. Pores volume was subtracted from the general volume. Knowing that Baltic sediments are mainly clastic, quartz density = 2.65 g/cm$^3$ (Anthony et al., 2009) was assumed to correspond to hypothetical 0%-porosity sediment. Combining modelled volume and density led to estimation of the overall mass of Baltic Sea Holocene sediment as well as annual rate of deposition averaged over the period.

### 3.6. Generation of paleogeographic maps

Having all the needed components described in Equation (1) and (2), namely the eustatic sea level change, the spatial distribution of the GIA and the sediment depositional thickness, the present-day DEM was transformed into a set of paleo-geographic maps for the Baltic Sea, mimicking the paleogeographic evolution of the region with fine temporal resolution (Supplementary Materials). The thickness of sediment deposition for each time slice was subtracted from the total Holocene thickness assuming constant sediment accumulation rate.

### 4. Results

### 4.1. Holocene sediment thickness map

Combination of 8 local-scale sediment thickness datasets and application of 3 extrapolation methods allowed to generate the regional Holocene sediment thickness map (Fig. 6). Several sites characterized by high sediment thickness were identified in southern and central part of the Baltic Sea, corresponding to depressions of sub-basins including the Arkona Basin, the Bornholm Basin as well as the Eastern and Western Gotland Basins. Maximum deposition thickness reaches up to 36 m in the Arkona Basin and the Eastern Gotland Basin. Although regions of enhanced sediment accumulation are usually



located in deeper basins, some shallower coastal areas also host locally-confined deposits with thickness larger than 20 m.
These include the central sections of the Gulf of Finland, the Gulf of Riga (both coupled with local bathymetry, as reported
by Jakobsson et al. (2019)) as well as the Gdańsk Basin. The thick Holocene deposit in the Gdańsk Basin is related to the
formation of the Hel Peninsula that is driven by alongshore sediment transport (Uścinowicz, 2022). In contrast to the
southern Baltic Sea, the Holocene sediment thickness in the northern Baltic Sea is relatively thin and mostly less than 6 m.
Such feature may be related to the glacio-isostatic uplift resulting in a continuous reduction of accommodation space in the
Bothnian Sea and the Bothnian Bay (Varela, 2015). The characteristics of seabed substrate of the northern Baltic Sea
confirm the presence of glacial clay, hard bottom complexes and bedrocks covered by a thin layer of Holocene sand and
gravel (Kaskela et al., 2012).

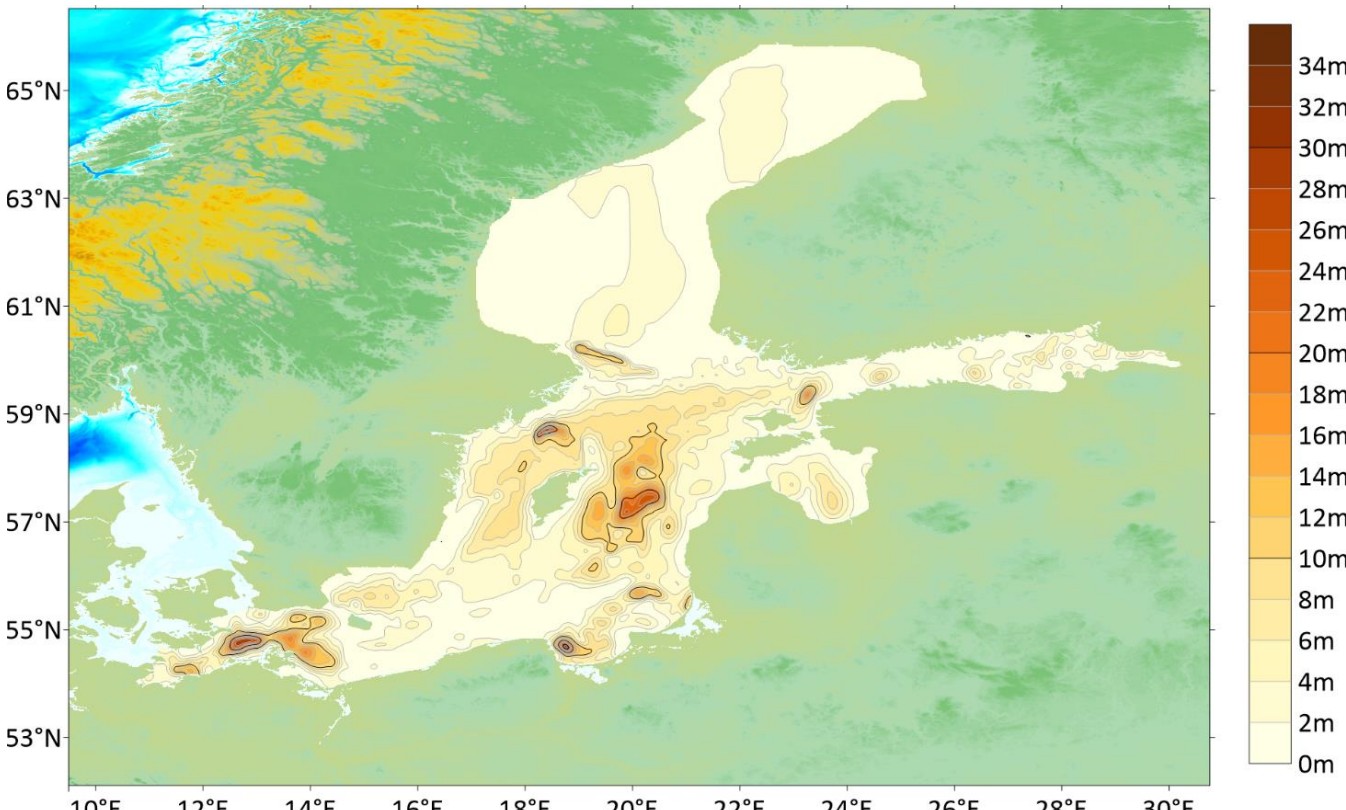

**Figure 6:** Regional Holocene thickness model derived from synthesis of 8 local sub-datasets and application of 3 extrapolation
methods.



## 4.2. Sediment budget

Sediment volume calculated using the trapezoidal rule, the Simpson's rule and the Simpson's 3/8 rule respectively provided nearly identical results (<0.001% difference, see Supplement). The calculated total volume of Holocene sediment is $1.372 \times 10^{12}$ m$^3$ in bulk. Subtracting the sediment porosity yields a zero-porosity sediment volume of $5.07 \times 10^{11}$ m$^3$,

corresponding to a sediment mass of $1.34 \times 10^{12}$ t and an annual sediment accumulation of $1.15 \times 10^8$ t yr$^{-1}$ in the present-day Baltic Sea.

## 4.3. Paleogeographic maps

A set of paleogeographic maps with a time interval of 500 years reflecting the evolution of the Baltic Sea region during the Holocene is provided in the Supplementary Materials, with maps for several periods marking a critical transition

in the sea level change shown in Fig.7. To evaluate the paleogeographic maps, a reference is made to the widely used maps derived from proxy data interpretation (Fig. 1) produced by Andren et al. (2010).

Starting at ~11.7 kyr BP, a transition from the Baltic Ice Lake (BIL) to the Yoldia Sea stage occurred. At that time, a large part of the north of today's Baltic Sea basin was covered by the glaciers of the Fennoscandian Ice Sheet (FIS). The water level of the Baltic Ice Reservoir was dammed up to 25 m above the present-day sea level (Andren et al., 2011) before the

lake water made its way through the Central Swedish Depression in the southern border of the FIS and flowed into the paleo-North Sea basin in a drainage event in the area of today's Kattegat (Fig. 7a). Rising sea level outpacing the moderate uplift of the mainland in the immediate vicinity of south of the ice margin in the Central Swedish Depression sustained a free exchange of water through a gate between the Baltic Sea and the North Sea for several centuries that marks a brackish-marine phase so-called the Yoldia Sea Stage (11.7 –11.0 kyr BP). Both models, the map shown in Fig. 7b and Andren's et al.

(2011) (Fig. 1b) results coincide in the existence of such a gate. The maps clearly show the course of the inherent BIL drainage in the lowlands of the Central Swedish Depression through lakes Vänern and Vättern. This drainage course was near the margin of the FIS. According to Patton et al. (2017), Shaw et al. (2006) and Stokes et al., (2015), the continental ice front is not to be consered a stable line but consists of multiple glacial-fluvial channels between large ice blocks. The central Swedish lowlands were characterized by such an environment, enabling the water flow between the Baltic Sea and the North

Sea during the Yoldia Sea period (Fig. 7a and 7b). During the Yoldia Sea stage (Fig. 7b) the sea level and the lake level in the Baltic Sea basin converged, which lasted until ~11.0 kyr BP. As the ice front continued to retreat northwards, the increasing uplift of Scandinavia outpaced the eustatic sea level rise and closed the gate in the central Swedish lowlands. The Baltic Sea consequently reverted to a freshwater environment, namely the Ancylus Lake, fed primarily by meltwater from the remnants of the FIS, which still covered the highlands of Scandinavia. The increasing flow of meltwater into the Ancylus

Basin led to a permanent rise in lake level (so-called "Ancylus Transgression") having reached its maximum at ~10.5 kyr BP and a subsequent drainage of the lake water into the paleo-North Sea (Fig. 7c). Different from the drainage through the



central Swedish lowlands in the later stage of the BIL, the freshwater outflow in the late stage of the Ancylus Lake moved to the south due to the increasing glacio-isostatic uplift of Scandinavia and took place through the area of today's western Baltic Sea, namely the Belt and the Sound. In the generation of the maps, we distinguished the water level between the Lake

Ancylus and the open North Sea from 11.0 to 9.5 kyr BP according to Uścinowicz (2006), which are shown in Fig. 3.

At around 8.0 kyr BP, the one-way drainage from the Baltic Sea (Lake Ancylus) to the North Sea ceased due to a rise of the eustatic sea level which caught up with the water level in the Lake Ancylus (Harff et al. 2017), and the Littorina transgression enabled a free exchange between the Baltic Sea and the open North Sea again (Fig. 7d). Since then, the global sea-level change has dominated the hydrographic regime in the Baltic Sea. Glacio-isostatic movements of the region lead to

persistent transgression in the south and regression in the north of the Baltic Sea (Harff et al. 2017). The paleogeographic setting at 6.5 kyr BP is illustrated by the map shown in Fig. 7d. A comparison with the map generated according to proxy data in Fig. 1c according to Andren et al. (2011) shows a general agreement both in the course of the coasts and in the connections of the Baltic Sea basin to the paleo-North Sea and contributes to the verification of the modeling undertaken here.


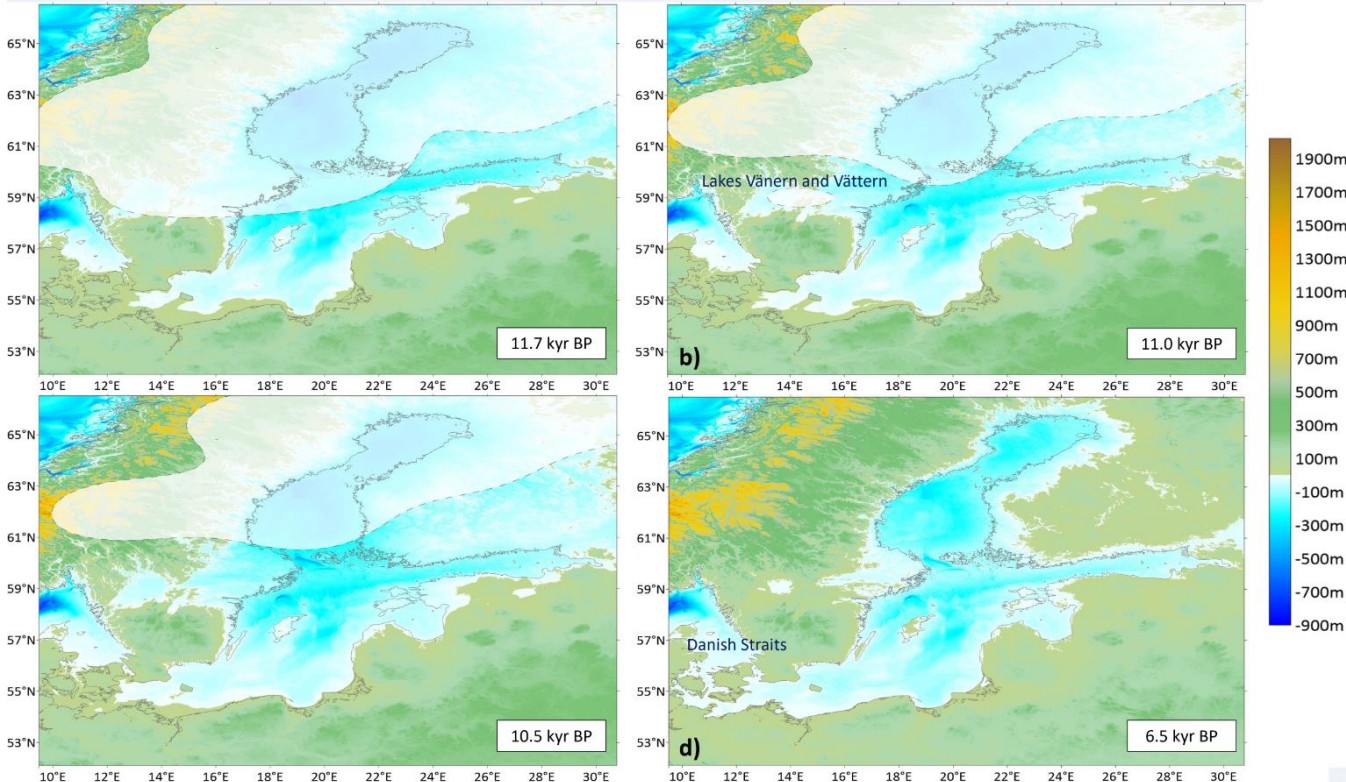

**Figure 7:** Reconstructed paleogeographic maps representing stages of the Baltic Sea. Four exemplary maps are depicted here for comparison with the paleogeographic maps generated by Andren et al., 2011. a) Baltic Ice Lake (just prior to the final





drainage), 11.7; b) Yoldia Sea (end of the brackish phase), 11.0 kyr BP; c) Ancylus Lake (transgression maximum), 10.5 kyr

BP; d) Littorina Sea (most saline phase), 6.5 kyr BP. Gray line depicts the present-day coastline.

## 4.4 Validation of the paleogeographic reconstruction

An interplay between eustasy and spatially varied rates of isostatic rebound is mirrored in various local RSL curves. Reconstructed curves for all subareas of the Baltic Sea Basin gathered by Rosentau et al. (2021) allowed to locally validate the numerically modelled paleogeographic maps. All curves located along eastern (Berglund, 2012; Hansson et al., 2018)

and northern Swedish coast (Linden et al., 2006) as well as all Finnish datasets (Glückert, 1976; Saarnisto, 1981; Mietinen, 2003) infer a decreasing RSL trend, dominated by isostatic uplift, whereas stations of southern Baltic mirror fast RSL rise that starts to slow down and stabilize after 6.0 kyr BP (Lampe et al., 2004, 2011; Gelumbauskaite, 2009; Uścinowicz et al., 2011; Miotk-Szpiganowicz et al., 2013,). Stations along Latvian and Estonian coast (Grudzińska et al., 2013, 2014; Lougas, and Tomek, 2013; Berzins et al., 2016) infer a decreasing trend distorted by the Ancylus transgression. Described trends

were well-mirrored in obtained maps (Fig.8). The opposite patterns of the RSL change between the northern and southern coasts gradually converge towards the central Baltic area where eustatic and isostatic components neutralize each other. This is seen at the stations in Latvia and Blekinge which show a relatively flat RSL curve with small-scale up-and-down variations (Damusyte, 2011;Rosentau et al., 2013; Habicht et al., 2017). Such relatively stable RSL is reproduced in our results, albeit with fluctuations in the period between 11.7 and 10 kyr BP. Comparison of the obtained paleogeographic

scenarios fit relatively well both regional field-observation-based reconstructions as well as local RSL curves (Fig.8), which validates the applied paleogeographic modeling methodology.





**Figure 8.** Comparison of the modelled RSL (red curves) and local RSL curves (black curves) derived from field data compiled by Rosentau et al. (2021).




## 5. Discussion

### 5.1. Comparison with existing reconstructions

The maps generated in this study are further assessed by comparison with existing field-based reconstructions. The comparison of proxy-data based RSL black curves published by Rosentau et al. (2021) and the model driven red curves shows a well interpretable similarities. The curves from northern Finland represent a straight sea-level fall because of the dominating regional GIA uplift of the regional Earth's crust, whereas the Vistula Spit curves shows continuous sea-level fall caused by additive effect of eustatic sea-level rise and local GIA controlled land subsidence because of the collapsing lithospheric bulge. The Blekinge curve displays relative stable conditions closed to the isostatically neutral hinge line that separates the "uplifting Baltic North" from the "subsiding Baltic South". At the Latvian west coast, the model data show the effect of damming during the Ancylus Lage stage up to ca. 10 cal. kyr BP and afterwards the sea-level drop caused by the lake drainage.

To aid the comparison, the coastline and the edge of the ice cover from regional paleogeographic reconstructions provided by Andrén et al. (2011), which have been widely referred to in existing literature, were overlaid with our maps (Fig. 9). The map shown in Fig. 9a represents the scenario for 11.7 kyr BP, corresponding to the beginning of the Holocene and the Baltic Ice Lake/Yoldia Sea transition. General paleogeographic features are consistent between our map and the one from Andrén et al. (2011). The Danish straits are closed at that time, whereas a connection between the Baltic Sea and the North Sea is being formed throughout the Central Swedish Lowlands. A major difference is on the morphology of Bornholm Island and Gotland Island. In Andrén's reconstruction, both islands are emerged already in 11.7 kyr BP. By contrast, our map shows that the Island of Bornholm is still connected to the mainland and Gotland Island is largely submerged at that time. The shape of the coastline remains similar, however depicts higher RSL in the southern part and lower RSL in the western and eastern parts of the Baltic Sea in Andrén's reconstruction compared to our result. Fig. 9b corresponds to the end of Yoldia Sea phase at 11.0 kyr BP. In both reconstructions the connection between the Baltic Sea and the North Sea via the Central Swedish Lowlands is being closed, and the Bornholm Island is connected to the mainland. The maximum of Ancylus transgression at 10.5 kyr BP is depicted in Fig.9c. In both reconstructions the North Sea and the Baltic Sea are disconnected. Coastlines in the southern part are similar between two maps. However, the difference increases northwards, inferring a lower RSL leading to emerging of West Estonian archipelago and Finland in Andrén's reconstruction compared to our result. Noteworthy is, that even though the difference in area seems to be vast, the difference in the elevation is relatively small, suggesting that the land-sea transition in this part is highly sensitive to the change of RSL. Our reconstructed morphology of the Littorina Sea at 6.5 kyr BP is characterized by an open connection between the Baltic Sea and the North Sea through the Danish Straits, which persists until today. In the reconstruction by Andrén et al. (2011), the Baltic-North Sea connection at 6.5 kyr BP exists only via the Great Belt, whereas all three straits are already opened in our result at that time. Similar to earlier scenarios, the RSL in the northern and central parts of the Baltic Sea is generally lower in the map of Andrén et al.





(2011) compared to our result. In summary, major features of the paleogeographic evolution of the Baltic Sea are well mirrored and consistent in both reconstructions. Differences are mainly related to the location of the paleo-coastline due to
higher RSL in the central and northern parts and lower RSL in the southern part of the Baltic Sea in our result compared to that of Andrén et al. (2011), which also affects the timing for the emergence of the islands.

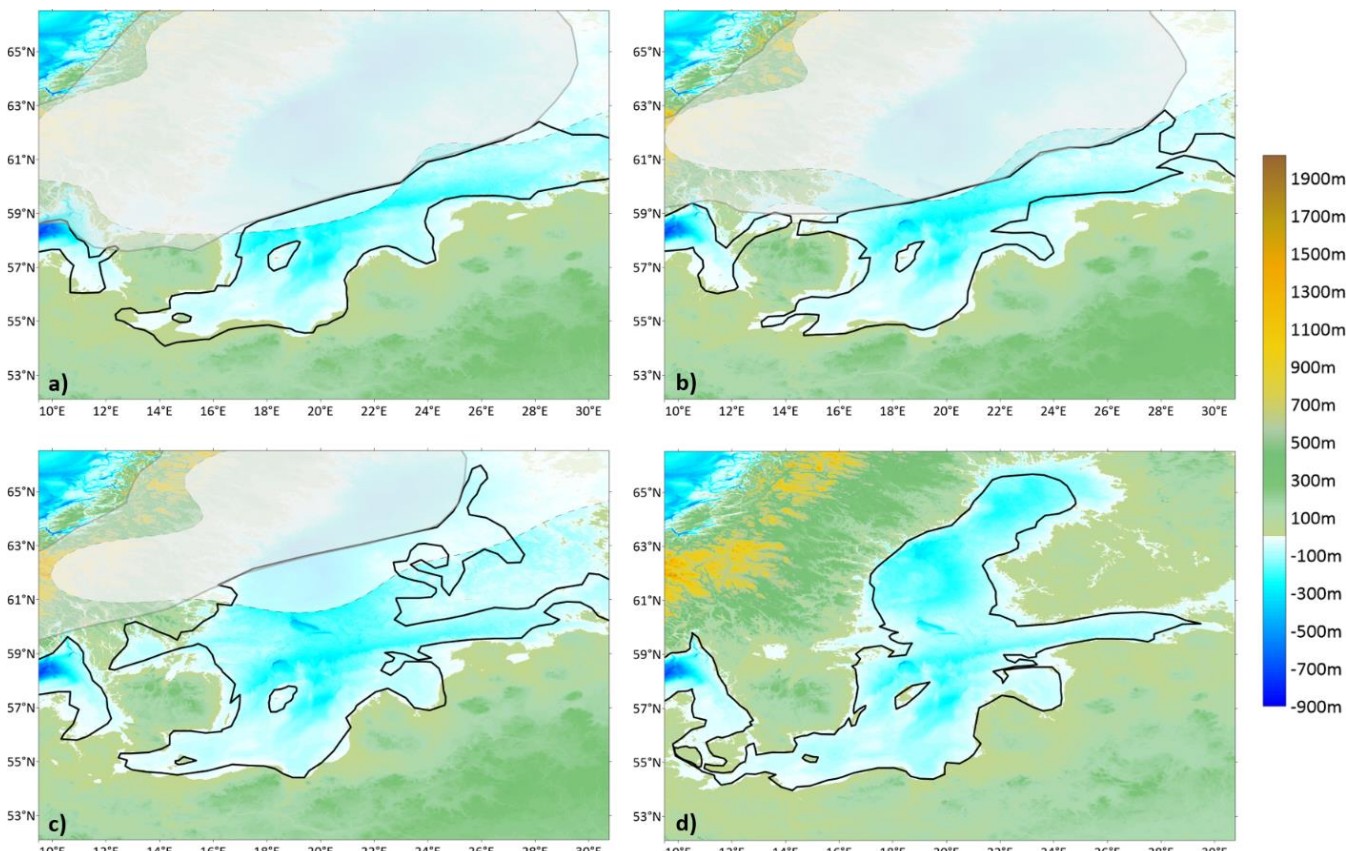

**Figure 9:** Reconstructed paleogeographic maps representing stages of the Baltic Sea overlaid with a paleo-coastline from
Andrén et al. (2011) for comparison: a) Baltic Ice Lake (just prior to the final drainage), 11.7 kyr BP; b) Yoldia Sea (end of the brackish phase), 11.0 kyr BP; c) Ancylus Lake (transgression maximum), 10.5 kyr BP; d) Littorina Sea (most saline phase), 6.5 kyr BP. Black bold line depicts a paleo-coastline whereas greyish polygon stands for ice extent after Andren et al. (2011).

### 5.2. Holocene sediment budget in the Baltic Sea

Our calculated sediment mass of the Holocene deposit in the Baltic Sea is $1.34 \times 10^{12}$ t, corresponding to an annual
sediment accumulation rate of $1.15 \times 10^{8}$ t yr$^{-1}$. Potential errors in the estimation may originate from several sources, including porosity, grain density and vertical compaction. The standard deviation σ of porosity is ~0.15 according to the



sediment samples analyzed by Endler et al. (2015). Applying the mean $\pm \sigma$ in the porosity data as the upper and lower estimates respectively indicates a range between $0.82\times10^{12}$ t and $1.87\times10^{12}$ t in the total sediment mass, corresponding to annual sediment accumulation rate between $0.69\times10^8$ t yr$^{-1}$ and $1.61\times10^8$ t yr$^{-1}$. Although the Holocene deposits mainly

consists of silt and sand, the deeper basins (e.g. Arkona, Bornholm and Gotland) are covered by a layer of mixture of organic matter (up to 16%) and clay (Leipe et al., 2011). The thickness of such layer can extend to several meters in the deep basins (Andrén et al., 2000; Ponomarenko, 2023). Assuming that the average thickness of this layer is 4 m (as indicated in a majority of sediment cores) in the basins, neglection of the vertical compaction results in an overestimation of the porosity by ~10% according to Schmedemann et al. (2008), corresponding to an underestimation of sediment mass of ~ $0.055\times10^{12}$ t,

which accounts for ~4% of the estimated mean total budget ($1.34\times10^{12}$ t).

A comparison of the annual sediment accumulation rate averaged over the Holocene with the present day's estimation by Porz et al. (2021) suggests that they are at the same order of magnitude for the SW Baltic Sea. The Holocene-averaged accumulation rate is $8.8\pm4.1\times10^6$ t yr$^{-1}$ in the SW Baltic Sea according to our data in this study. In the budget analysis done by Porz et al. (2021), the annual accumulation rate of fine-grained sediment in the SW Baltic Sea basins is

between $5.5\times10^6$ and $8.2\times10^6$ t yr$^{-1}$, with coastal erosion, namely erosion of the glacial-till cliffs, as the main source that contributes 30-50% of the annual deposition in the basins. River supply and biogenic production are an order of magnitude smaller than coastal erosion and account for ~5% and 4-15% of the annual deposition, respectively. In addition, sediment input from the North Sea is estimated to be on the order of $1\times10^6$ t yr$^{-1}$, contributing to 10-30% of the annual deposition in the SW Baltic Sea. Since during the Ancylus Lake stage the connection between the Baltic Sea and the North Sea was

closed, the sediment supply from the North Sea was halted for that period which lasted for ~1.5 kyrs. This implies a compensation of this gap by enhanced land supply of sediment e.g. from melting of the ice cover on the Scandinavia, which is larger than present-day values.

**5.3. Importance of integrating sediment dynamics in paleogeographic reconstructions**

Depending on the regional or local setting, the individual impact of eustatic, isostatic/tectonic and sediment

dynamics on geographical and morphological development of marginal seas may vary significantly. Inclusion of sediment dynamics in paleogeographic reconstruction of coastal regions and continental shelves that are fed by significant terrestrial sediment input has critical influence on the location of the paleo-coastline and general morphology of the seabed. This has been demonstrated in the paleo-reconstructions of the Beibu Gulf in the northern South China Sea (Xiong et al., 2020; Zhang et al., 2020), the Pearl River delta and its estuary (Wu et al., 2010), the Mekong River delta and adjacent shelf (Wang et al.,

2024), the southwestern coast of Bohai Sea (Liu et al., 2016) and the southern North Sea (Van der Molen and Van Dijck, 2000).

The integration of sediment dynamics is also important for understanding the evolution of large-scale sedimentary systems which are not directly fed by riverine sediment but are formed and/or shaped by sediment transport, such as barrier



islands (Zhang et al., 2011a; Zhang et al., 2014; Karle et al., 2021) and mud depocenters (Porz et al., 2021). This process is

especially important for evolution of the southern Baltic Sea where various barrier islands have developed since mid-Holocene when the sea level has approached a relatively stable level (Uścinowicz et al., 2011; Zhang et al., 2011b; Dudzińska-Nowak, 2017). It is worth noting that although many paleogeographic reconstructions exist at local scales by considering sediment transport dynamics, our work represents the first attempt for a consistent reconstruction at a marginal sea scale. Comprehensive information of dated sediment thickness for a marginal sea such as the Baltic Sea is difficult to

obtain, as most of relevant datasets are derived locally and it requires extensive effort in collection, integration, harmonization and synthesis of such datasets and filling of gaps between them. A consistent paleogeographic reconstruction of a marginal sea considering not only regional processes such as eustatic sea level change and isostatic/tectonic movement but also sediment transport provides indispensable information on the historical development of the marginal sea especially its coast. Such information is critical for assessment of the future state of marginal seacoast in response to climate change

and human impacts.

## 6. Conclusions

This study presents a spatially high-resolution (0.01°×0.01°) paleogeographic reconstruction of the Baltic Basin including the coastal evolution of the Baltic Sea for the Holocene based on the application of a conceptual equation to link empirical (primary) and model (secondary) data. These data describe the surface structure of the study area, climate-induced

eustatic sea level changes, as well as glacio-isostatic (GIA) vertical Earth's crust movement and the thickness evolution of sediment deposition in the Baltic Sea Basin starting with the end of the Baltic Ice Lake stage 11.7 kyr BP. Local datasets of sediment thickness from open sources including existing literature and data portals with public access were compiled and complemented by numerical inter- and extrapolations to generate a consistent regional map of Holocene sediment thickness for the Baltic Sea. The map shows that relatively thick Holocene sediments are deposited in the southern and central parts of

the Baltic Sea, filling sub-basins, including Arkona Basin, Bornholm Basin, Eastern and Western Gotland Basins and Northern Central Basin, with a maximum thickness of up to 36 m. In addition, some shallower coastal areas in the southern Baltic Sea also have localized deposits with a thickness of more than 20 m, mostly associated with alongshore sediment transport and formation of barrier islands and spits. In contrast to the southern Baltic Sea, the thickness of Holocene sediments in the northern Baltic Sea is relatively low, mostly less than 6 m. The total mass of Holocene sediment in the

Baltic Sea according to the thickness map and corresponding porosity data is estimated to be $1.34\pm0.53\times10^{12}$ t, which corresponds to a sediment accumulation of $1.15\pm0.46\times10^{8}$ t per year.

For the first time, the paleogeographic reconstruction of the Baltic Sea for the Holocene was achieved by a combination of crustal deformation (GIA), eustatic water level change and sediment accumulation considering the disconnection of paleo-North Sea and the easterly freshwater body during the Ancylus Lake stage. The model results and thus the functionality of

the model expressed by the conceptual equation is validated by comparison with field-based proxy data interpretation. This comparison improves the reconstruction of the hydrographic dynamics between Baltic and North Seas Basin, the marginal



zone of the former FIS. In the northern part of the Baltic Basin the differences in coastlines reconstructed by field (proxy) data and model results lead to the assumption that the model underestimated the GIA in the central areas of former FIS, so that model parameters should be improved. Our work thus represents a further step towards a consistent methodology to

reconstruct the formation of marginal seas during transgression/regression cycles including not only tropic and subtropic climate zones but also polar, subpolar marginal seas impacted by the regional dynamics of inland ice sheets. The comparison of model results with proxy-data interpretation allows the improvements of GIA model parameters. paleogeographic reconstruction on a marginal marine scale.

### Acknowledgements

This study is an outcome of the project "Morphological evolution of coastal seas – past and future" (https://marginalseas.ddeworld.org/margseas-rd-research-project) funded by the Deep-time Digital Earth program (https://www.ddeworld.org/). It is also supported by the Helmholtz PoF programme "The Changing Earth – Sustaining our Future" on its Topic 4: Coastal zones at a time of global change.

### Authors contribution


Wenyan Zhang designed and supervised the study. Jakub Miluch collected, digitized, and processed sediment data and eustatic sea level curve of the study area. He also generated the paleogeographic maps by numerical modeling supervised by Jan Harff. Andreas Groh provided the data and GIA scenarios used for paleogeographic modeling and contributed related text descriptions. Peter Arlinghaus applied the convolutional neural network to fill the data gaps in the sediment thickness.

Celine Denker assisted in collection, digitization, and generation of the sediment thickness map. Jakub Miluch and Wenyan Zhang wrote the original manuscript. All authors have contributed to manuscript revision.

### Data availability

Publicly available datasets were analyzed in this study. Present-day digital elevation model of the Baltic Sea is derived from

GEBCO 2023 Grid (doi:10.5285/f98b053b-0cbc-6c23-e053-6c86abc0af7b). Seismic profiles in the Gotland Basin were kindly provided by Dr. Peter Feldens from the Leibniz Institute for Baltic Sea Research (IOW). Gridded Holocene sediment thickness data produced in this study can be found at the Mendeley Data with doi: 10.17632/k45mff2ccy.1

### Competing interests

The contact author has declared that none of the authors has any competing interests.



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
