# Peer review of "Paleogeographic numerical modelling of marginal seas for the Holocene – an exemplary study of the Baltic Sea"

_EGUsphere, 2024_

## Referee Comment (RC2)

[referee-annotated manuscript omitted]

---

## Editor Comment (EC1)

General comments to the manuscript: Paleogeographic numerical modelling of marginal seas for the Holocene – an exemplary study of the Baltic Sea

Author(s): Jakub Miluch, Wenyan Zhang, Jan Harff, Andreas Groh, Peter Arlinghaus, and Celine Denker

The manuscript deals with paleogeography of the Baltic Sea during the Holocene by combining eustatic sea-level change, glacio-isostatic adjustment and sediment deposition.

They present paleoreconstruction and map of Holocene sediment thickness based on different datasets and calculate total mass of Holocene sediment in the Baltic Sea and yearly sediment accumulation.

Reading the manuscript there are terminological and methodological problems, which are described below.

First there are some errors using terminology, like using plate (plate tectonics) instead plain and its not clear what is meant by platform (lines 64-68 see comments below). For some terms it's not clear the meaning, like inland ice (possibly glacier), gate and gate function or amphibious Digital Elevation Model.

Secondly there are some methodological problems like Baltic Ice Lake /Yoldia Sea transition (look details below **Figure3**) and creation of sediment thickness map.

Chapter 3.4 **Sediment thickness** does not have information about the uncertainties of the used data sources. Why only present-day sea area data were used? In central and northern areas, like Gulf of Botnia, coastline was several hundred meters higher, and sedimentation in the Baltic Sea occurred also in present day mainland. Moreover referred Winterhalter 1972 does not have any datapoints from north of Gulf of Botnia (yellow square in Figure 5) so its not clear how those data were manipulated.

According to line 231 Holocene sediments in Southern Baltic are on top of glacial till. Holocene starts at 11.7 ka BP but glacial till accumulated around 17-15 ka BP, so there was no sedimentation several thousand of years? According to line 241 glacial varved sediments of the Baltic Ice Lake are considered early Holocene age, what is not true as Baltic Ice Lake drainage (end) coincides more or less with the start of Holocene, so Baltic Ice Lake sediments are from Pleistocene, not from Holocene. There has been earlier attempt to create Holocene sediment thickness by Jakobson et al 2007 https://doi.org/10.1016/j.gloplacha.2007.01.006, which differs from results here. So the map presented here seems to include not only Holocene sediments but some Pleistocene sediments also.

**There are some issues with Figures:**

**Figure 1** longitude values starting from 15° and specially 20°-40° are almost 5° wrong. Glacier extent specially for 10.5 ka BP is not the same as in Andren et al 2011. It seems that there is problem with georeferencing.

**Figure 2** according to figure Peltier 1999 ice thickness model was used (ICE-4G), but in text Peltier 2004 (ICE-5G)

**Figure 3** explains that authors have wrongly modelled Baltic Ice Lake at 11.7 ka BP or they don't understand how Yoldia Sea Stage started. According to that figure the highest BIL water level occurred 12 ka cal BP and not at 11.7 ka BP as suggested by Andren et al 2011. At 11.7 ka BP water level in BIL dropped during ca 1-2 years 25 meters and Yoldia Sea started. So modelled BIL at 11.7 ka BP (Fig 7) is actually Yoldia Sea first stage after BIL drainage not BIL prior the drainage.

**Figure 7** reconstruction for 11.7 kyr BP and 11.0 kyr BP look in the southern part near Bornholm exactly same

**Figure 8** the caption is not correct. Both curves red (results in manuscript )and black (Rosentau et 2021) are modelled RSL curves according to ICE-5G model. Rosentau et al 2021 has on Figures 7,8,9 shown results with ICE-5G model with three different litosphere thicknesses and also ICE-6G, which one is used here is not clear. Black curve is not field data (or proxy data). Only RSL for Finland N looks similar to Rosentau et al 2021 results.

**Figure 9** Comparison in present form is not convincing as shorelines from Andren et al 2011 seems to like freehand drawings and differ from original. Moreover, on Figure9 a) You compare results here with BIL prior final drainage (Andren et al 2011) but its water level was about 25 meter higher than in present reconstruction. That also explains why Figure 9a and 9b coastlines are so similar.

There are some spelling errors in references and reference list and not all reference area on the list. In reference list sometimes only first author is shown are not.

In the following are some comments by line numbers.

Chapter 1. **Introduction**

45 climatically controlled eustatic **sea level** changes

46 Lambeck 2010 not in reference list

Chapter **Geological setting**   some terminology is not clear/correct

62-63 **Danish Straits** and **Swedish Sound**??  First term is enough as it includes all straits, second is The Sound or Öresund

64,68  Russian Plate  - there is no such plate, do You mean East European (Russian) Plain?

65 what are **Western** and **Eastern European Platform**?

78, 85, 99, 143 the term **inland ice**   should be replaced by glacier/icesheet

82. **sea-level drop** What about if land uplift is smaller than sea level rise?

83,84 not clear what is meant by **gate and gate function**   its more like technical term used in artificial reservoirs not for natural waterbodies

85 That sentence is not clear and not correct as there are surely sediments and proxy data older than post-glacial period (=last 11700 cal yr BP)

89 There is no LGM on Fig. 1

100 Heinsalu and Veski were using brackish-water Yoldia Sea

102 why so-called?

110. Figure 1 longitude values are wrong. Maps are difficult to read, because its not clear what is light blue and what is blue in Gulf of Botnia and near Oslo fjord. Baltic Ice Lake existed in Pleistocene

Chapter **3. Data and methods**

144 the sentence meaning not clear

150 Figure 2  What ice model was used?

Chapter **3.2. Eustatic data (EC)**

166 Waelbroeck et al 2002 not in reference list

168 global ocean? what in none global ocean?

Chapter **3.3. Vertical crustal movements...**

191-195 add here some references

205 explain how You get 500 years timeslices for reconstructions if GIA resolution is 1000 years

Chapter **3.4 Sediment thickness**

That full section needs more explanation and some information about reliability and resolution of used data sources.

Chapter **Discussion**

399 Rosentau et al 2021  black curves are not proxy-based but modelled by ICE-6G_C(VM5a)

435  Figure 9 Andren et al 2011 coastlines are not similar to published maps.

439-440  that was already in chapter 4.2

---

## Author Comment (AC1)

**Documentation of changes and reply to the review comments**

*[The original review comments are in **bold and italic**]*

***REVIEWER COMMENTS:***

**Referee 1**

*This manuscript presents a marginal sea basin-scale reconstruction of palaeogeography based on the combination of a big dataset including the eustatic sea levels, the GIA and the Holocene sediment accumulation for the Baltic Basin. A comparison between the modelled results and proxy-data-based reconstruction can improve the GIA model parameters, which is important for the study on Holocene sea-level history. I only have minor comments as below:*

*1. Line 130, please explain why ΔSED should be included in the equation of relative sea level.*

Author response #1:

Sorry, it was our mistake in the writing process. ΔSED should not be included into eq. (1). This term was not included in our reconstruction procedure.

We have removed it from the revised version.

*2. line 305, constant sediment accumulation rate was assumed in the estimation of sedimentation thickness at each time slice. However, sedimentation rate varies largely because of the changes in relative sea level, sediment supply, etc. particularly in coastal zone. Authors made a discussion in the last section of 5.3 for effect of the sediment dynamics. I feel this is not enough and expect an evaluation of the spatial distribution of areas characterized by changing sedimentation rate.*

Author response #2:

We are aware of the fact that the sediment accumulation rates varied throughout the Holocene. Even though these rates could be estimated based on analysis of sediment cores (being point data), expanding it to other sections of Baltic Sea basin would be difficult to justify as sedimentary environments of the Baltic vary not only in time but also in space. Moreover in the Baltic Sea, the magnitude of ΔSED remains relatively small compared to the magnitudes of ΔEC and ΔGIA. Also, the highest sedimentation rates are situated in the deeper basins. Therefore assuming different sedimentation rates would have only minor influence on the paleo-bathymetry. We will add discussion for an evaluation of the uncertainty related to changing sedimentation rate to the revised version based on the above-mentioned arguments.

It is worth to note that our attempt is one of the first complex applications integrating ΔSED to paleogeographic reconstructions at a marginal sea scale. As pointed out in discussion, applying it to more sedimentation-dominated environment, such as i.e. SE Asian shelf, would require slightly different approach by taking into account sediment compaction, erosion and different accumulation rates.

*3. line 401, "whereas the Vistula Spit curves shows continuous sea-level fall" should be "whereas the Vistula Spit curves show continuous sea-level rise".*

Author response #3:

This mispelling will be corrected. Thank you for pointing it out.

---

## Author Comment (AC2)

**egusphere-2024-1931**

**Documentation of changes and reply to the review comments**

*[The original review comments are in **bold and italic**]*

***REVIEWER COMMENTS:***

**Referee 2**
*General comments to the submitted manuscript: 'Paleogeographic numerical modelling of marginal seas for the Holocene – an exemplary study of the Baltic Sea' by Miluch et al.*

*The paper describes the paleogeographic distribution of the Baltic Sea during the Holocene. They base their observations on different data sets gathered from different countries along the Baltic Sea. For the reconstruction, they mainly use three parameters: vertical changes of the water level, vertical changes of the landmasses (tectonics and isostasy), and the thickness of the sediment deposits.*

*The novelty is that they include the latter one. The paper is well structured, and the Figures are certainly relevant for the message.*

*After reading the manuscript I have some general discussion points left:*

1. *The spatial resolution of the grid size in the Baltic is not always clear. I have no idea how accurate and variable for instance the input data of the isostacy is, nor of those of the sediment thickness data. In qualitative sense, they indicate for instance at the data in the northern Baltic Sea is sparse, but I have no idea how sparse (in a quantitative sense, how many data points per km$^2$).*

Author response #1:

The grid size is uniform at 0.01° x 0.01°. All point data are interpolated to this grid. The initial digital elevation model derived from the latest product of GEBCO has the resolution of 15 arc seconds (0.004167° x 0.004167°). The GIA dataset was initially at a global-scale with 0.5° x 0.5° grid size. It was then extracted for northern Europe and re-gridded to the 0.01° x 0.01° grid.

Sediment thickness data were interpolated based on several original datasests from difference sources and with different spatial resolution, including gridded data for local parts, seismic profiles and point measurements. These source data were used for trainning and validation for the machine learning approach (U-net) which was used to fill the data gaps (extrapolation) in other parts of the study area. In addition, co-krigging method was also applied for interpolation at some local parts where existing measurements show high correlation between sediment type/water depth and sediment thickness.

We will add information about the resolution of each sub-dataset as a new table in the revised version for clarification.

2. *The spatial interpolation and extrapolation techniques are not always explicitly mentioned. A Table for all parameters including it's resolution and spatial interpolation technique would be nice.*

Author response #2:

We agree. A table for information about the resolution of each sub-dataset as well as the interpolation and extrapolocation techniques will be provided in the revised version for clarification.

> **3. The discussion about the different phases is much based on the glacial history of the area. While reading the paper, I wondered how large the influx from major rivers in central Europe was, and how this differed during the Holocene. These rivers will certainly transport sediments towards the southern shores of the Baltic.**

Author response #3:

In the paleogeographic reconstruction, we did not distinguish the contribution between riverine discharge and coastal erosion (mainly glacial till cliffs) in the total sediment deposition thickness. We did discuss the impact of rivers, though they were not the focus of this paper. As discussed in chapter 5.2 the riverine sediment supply rate is one order of magnitude smaller than the supply from soft cliff erosion in the southern Baltic. The assessment is based on present day's sediment discharge rate from major rivers including the Vistula River and Oder River. To make the discussion more comprehensible, we will add more descriptions on a comparison between riverine sediment inputs and coastal erosion.

> **4. The interpretation around the Danish straits is based on very few data and probably also modeled in a very course sense. The authors discuss their results with those over earlier studies and conclude that not all studies for instance have two straits. How comparable are these studies if both studies are based on not a lot of observations, nor have a detailed modeling grid.**

Author response #4:

The timing of events such as opening of straits varies between reconstructions. Some local-scale variabilities of narrow structures such as straits may not be well visible on the regional maps due to coarse map resolution, difference in the data coverage and/or reconstruction methods. Therefore we agree that it is not comparable between regional studies on local-scale features which do not have sufficient data coverage. We will modify the discussion on this aspect.

> **5. The novelty is concentrated on the inclusion of sediment thickness data for the Baltic Sea. In the manuscript, the authors talk about 'sediment dynamics', 'sediment budget', 'sediment mass', and 'sediment thickness'. The paper becomes much more precise when a clear terminology is chosen. Really like your terms 'thickness' or 'mass'. Perhaps even better 'thickness' and 'vertical changes in thickness'. 'Budget' is perhaps more than you present (with a budget, you normally also mention lateral fluxes in the system). 'Dynamics' is closely related to sediment transport resulting in areas with erosion or deposition and you are not showing that.**

Author response #5:

We refer to different meanings when using different terminologies. 'Sediment dynamics' corresponds to dynamic processes including sediment transport, deposition and erosion. We used this term to highlight the novelty of our method by including the impact of sediment deposition on the paleo-morphology and explain why integrating sediment dynamics is important for paleogeographic research. 'Sediment mass' is an overall mass of sediments calculated from a grid, whereas 'sediment thickness' refers to the vertical thickness of sediment deposition for the period of interest. 'Sediment budget' refers to detailed analysis regarding sediment sources, sinks and transport pathways. Clarification of each term will be added in the revised manuscript. We agree that lateral fluxes such as riverine input, coastal erosion and sediment exchange between the North Sea and the Baltic Sea should be included in the sediment budget analysis, and we will add estimates of these fluxes in the revised version.

6. **More specific comments are included in the pdf-file of the manuscript.**

Author response #6:

Thank you for the attached comments. We have implemented modifications to text based on the comments. Here a couple of answers to your important questions or comments:

Line 42. We decided to leave 'post glacial period' here. We believe it's clear enough and it's not strictly limited to very beginning of Holocene.

Line 56. Above we discuss factors in general. Here we discuss details of model components.

Lines 146 and 147. These sentence discusses in general tools of the software, which comprises of various interpolation methods and allows to generate grids with pre-defined resolution.

Line 172. It is not specified however we know that this region, due to being situated on the hinge-line is the least-GIA-influenced area of the Baltic.

Line 278. It's purely random. This method is characterized by little randomness.

Line 299. Sediment budget analysis, discussing the sediment sources is conducted in chapter 4.2.

Line 305. We are aware of the fact that the sediment accumulation rates varied throughout the Holocene. Even though these rates could be estimated based on analysis of sediment cores (being point data), expanding it to other sections of Baltic Sea basin would be hard to justify as sedimentary environments of the Baltic varied not only in time but also in space. Moreover at the Baltic Sea regional scale, ΔSED remains as a minor component of the equation in comparison to ΔEC and ΔGIA. Also, the highest sedimentation rates are situated in the deeper basins. Therefore assuming different sedimentation rates would have very minor influcence on the paleo-bathymetry.

Our attempt is one of the first applications of this method. As pointed out in discussion chapter, applying it to more sedimentation-dominated environments, such as the SE Asian shelf, would require a slightly different approach by distinguishing the impact of sediment erosion and accumulation rate in different periods. However, on the other hand, incorporating complicated paleo-morphodynamic models of sediment accumulation, erosion, redeposition or compaction as well as sediment fluxes will bear high uncertainty due to insufficient data for model forcing, boundary configuration , calibration and validation. This needs to be addressed in future studies.

Line 369. These locations are the connections between the Baltic and the North Sea. We refer to them in the text, therefore we decided to place them on the map.

Line 436. Indeed. Paleo-cliffs may be found inland of Wolin Island (Poland) nearby Szczecin Lagoon shoreline (it was a marine bay during Littorina trangression). Also terraces of paleo-coastlines are well-seen on the Gotland Island, however, in this case it is mainly related to isostatic uplift.

Line 457. In this paragaph we simply compare present sediment accumulation rates from Porz et al. (2021) with our modelled thickness and refer to the proportion of sediment sources input to the basin.

Line 463. We would rather leave the word „dynamics" here. Even though for the Baltic Sea study we simply subtract the thickness grid, the following chapter acts as an outlook for future studies. We discuss application of this method to more sedimentation-dominated environments in which

sediment dynamic processes (erosion, accumulation, redeposition, sediment fluxes) need to be taken into account, or application of it in a local scale.

Line 467. There are spits like this in the Baltic coast: Hel Peninsula (Poland), Vistula Spit (Poland/Russia) or Curonian Spit (Russia/Lithuania). We would rather keep the sentence and add examples into the text.

Line 509. In this sentence we summarize what is a new achievement of our study and discuss application of consistent methodology of paleogeographic modeling into different climatic zones. There are no new material here.

---

## Author Comment (AC5)

**egusphere-2024-1931**

**Documentation of changes and reply to the review comments**

*[The original review comments are in **bold and italic**]*

*REVIEWER COMMENTS:*

Referee 3

1.  **General comments to the manuscript.**
    **The manuscript deals with paleogeography of the Baltic Sea during the Holocene by combining eustatic sea-level change, glacio-isostatic adjustment and sediment deposition.**

    **They present paleoreconstruction and map of Holocene sediment thickness based on different datasets and calculate total mass of Holocene sediment in the Baltic Sea and yearly sediment accumulation.**

    **Reading the manuscript there are terminological and methodological problems, which are described below.**

    **First there are some errors using terminology, like using plate (plate tectonics) instead plain and its not clear what is meant by platform (lines 64-68 see comments below). For some terms it's not clear the meaning, like inland ice (possibly glacier), gate and gate function or amphibious Digital Elevation Model.**

Author response #1:

Thank you for the comment. We are aware of terminological discrepancies in structure description Northeast and Southwest of the SST/TTZ suture zone. We will change the terminology according to definitions given by Uscinowicz (2014) and Maystrenko et al. (2008).

We deliberately chose the term "gate" to describe the hydrographic interaction of sea level rise and vertical coastal movement in opening and closing the connecting routes between the Baltic Sea basin and the Paleo-North Sea. This term is widely used in literature for such topographic features. As you point out below „gate function" sounds more like a technical term. This is why we put it in „quotation marks".

„Inland ice" stands here indeed for glacier. We will change the term accordingly. „Amphibiuous Digital Elevation Model" refers to a DEM that covers both subaquous and subaerial parts. We will add an explanation and remove the word „amphibiuous" to avoid confusion.

2.  **Secondly there are some methodological problems like Baltic Ice Lake /Yoldia Sea transition (look details below Figure3) and creation of sediment thickness map.**
    **Chapter 3.4 Sediment thickness does not have information about the uncertainties of the used data sources. Why only present-day sea area data were used? In central and northern areas, like Gulf of Botnia, coastline was several hundred meters higher, and sedimentation in the Baltic Sea occurred also in present day mainland. Moreover referred Winterhalter 1972 does not have any datapoints from north of Gulf of Botnia (yellow square in Figure 5) so its not clear how those data were manipulated.**

Author response #2:

Information about uncertainties and input data resolution will be added to the manuscript (see also Author response #4). The idea of generation of „Baltic Holocene sediment thickness" map was to limit it to present-day Baltic Sea. Regarding sedimentation on present-day mainland (mainly the northern Baltic coast) after it was emerged, this part of sedimentation has been subjected to reworking by both water flow and wind, therefore any estimation of marine sediment thickness would be even more difficult to obtain and justify. Moreover, the Holocene sediment thickness of the northern Baltic Sea and its coast is generally very thin that any extrapolation to the mainland would have no visual impact on the paleo-DEM and very minor impact on the total estimated sediment budget of the Holocene. We will include discussion of this aspect in the revised manuscript.

Regarding Winterhalter 1972 dataset it was indeed limited to the southern Gulf of Bothnia. Unfortunately we could not find any literature providing sedimentation thickness in the northern Gulf of Bothnia. This is why we used co-kriging to extrapolate it to the north (explained in lines 256-262). A brief explanation will be added for clarification.

3. **According to line 231, Holocene sediments in the southern Baltic Sea are on top of the glacial till. Holocene started at ~11.7 ka BP but glacial till accumulated around 17-15 ka BP, so there was no sedimentation several thousand of years? According to line 241 glacial varved sediments of the Baltic Ice Lake are considered early Holocene age, what is not true as Baltic Ice Lake drainage (end) coincides more or less with the start of Holocene, so Baltic Ice Lake sediments are from Pleistocene, not from Holocene.**

Author response #3:

Top of glacial till is a well-visible seismic reflector, whereas border between Baltic Ice Lake (BIL) and Yoldia Sea is hard to determine from the seismic profiles. Due to fact that BIL sediment chronostratigraphically belongs to Late Pleistocene, the modelled thickness may be slightly overestimated. We will add an explanation to chapter 3.4.2 including an estimate of uncertainty related to this.

4. **There has been earlier attempt to create Holocene sediment thickness by Jakobson et al 2007 https://doi.org/10.1016/j.gloplacha.2007.01.006, which differs from results here. So the map presented here seems to include not only Holocene sediments but some Pleistocene sediments also.**

Author response #4:

According to the information from Jakobson et al. (2007), the Holocene sediment thickness map for the Baltic Basins shown in their Figure 3d was compiled through assembling information from available sediment distribution maps and information retrieved from the Swedish Geological Survey's mapping archives which unfortunately do not provide an open access. A comparison between our map (Figure 5) and the map from Jakobson et al (2007) shows a general agreement in the Borholm basin and along the Swedish coast near the Gotland. However, there exists a large descrepency in the thickness value in other basins (e.g. Arkona basin, Gotland basin) between the two maps. The thickness values in Jakobson et al. (2007) for these basins are much smaller than previous published values from Lemke (1998) and Uścinowicz (1998) focusing on these local areas. The compiled thickness data from the difference sources we have collected show more consistent patterns covering both deep basins and shallow coastal areas and therefore we argue that our map provides a more accurate distribution of Holocene sediment thickness compared to earlier publications. We will add discussion on this aspect in the revised version.

5. **There are some issues with Figures:**

   **Figure 1 longitude values starting from 15° and specially 20°-40° are almost 5° wrong. Glacier extent specially for 10.5 ka BP is not the same as in Andren et al 2011. It seems that there is problem with georeferencing.**

Author response #5:

Thanks for pointing this out. We admit that it is a mistake in our georeferencing. We will provide updated figure to correct this problem.

6. **Figure 2 according to figure Peltier 1999 ice thickness model was used (ICE-4G), but in text Peltier 2004 (ICE-5G)**

Author response #6:

It will be corrected to Peltier (2004) in Fig. 2.

7. **Figure 3 explains that authors have wrongly modelled Baltic Ice Lake at 11.7 ka BP or they don'tunderstand how Yoldia Sea Stage started. According to that figure the highest BIL water level occurred 12 ka cal BP and not at 11.7 ka BP as suggested by Andren et al 2011. At 11.7 ka BP water level in BIL dropped during ca 1-2 years 25 meters and Yoldia Sea started. So modelled BIL at 11.7 ka BP (Fig 7) is actually Yoldia Sea first stage after BIL drainage not BIL prior the drainage. In the 11.7 kyr BP map by Andren (2011), the hydrographic connection between the Baltic Sea basin and the Kattegat indicates communicating systems. This makes Andren's caption "(...) BIL prior to final drainage" questionable.**

Author response #7:

Please note that the red line on Fig 3. stands for the Uścinowicz (2006) RSL curve that we used only for the Ancylus lake phase. Our reconstruction started exactly at 11.7 kyr BP when the water level of the Baltic region was dropped to the level of the open North Sea. You are right that modelled BIL at 11.7 kyr BP is actually Yoldia Sea first stage after BIL drainage but not BIL prior the drainage. We will correct „BIL prior to drainage" to „Baltic Ice Lake/Yoldia Sea transition" in all relevant texts including captions of Figure 1 and 7.

8. **Figure 7 reconstruction for 11.7 kyr BP and 11.0 kyr BP look in the southern part near Bornholm exactly same**

Author response #8:

They are similar but not the same. The RSL on the map 7b is higher than that on map 7a what makes the land bridge connecting Bornholm with mainland narrower. We will add a brief explanation on this in the text.

9. **Figure 8 the caption is not correct. Both curves red (results in manuscript )and black (Rosentau et 2021) are modelled RSL curves according to ICE-5G model. Rosentau et al 2021 has on Figures 7,8,9 shown results with ICE-5G model with three different lithosphere thicknesses and also ICE-6G, which one is used here is not clear. Black curve is not field data (or proxy data). Only RSL for Finland N looks similar to Rosentau et al 2021 results.**

Author response #9:

Thank you for pointing it out. We will correct Fig. 8 caption and remove the word „field data".

Regarding the model, we digitized ICE-5G model with 120km lithosphere thickness curves. We will add this information to the caption. Putting all four curves overlaid with ours would have been too much information on one graph. We agree that the RSL of Rosentau et al (2021) show differences with our results mainly for the initial stage between 11.7- 10 kyr BP and afterward a general agreement is reached. We will add some descriptions and discussion on this aspect in the revised version.

10. **Figure 9 Comparison in present form is not convincing as shorelines from Andren et al 2011 seems to like freehand drawings and differ from original. Moreover, on Figure9 a) You compare results here with BIL prior final drainage (Andren et al 2011) but its water level was about 25 meter higher than in present reconstruction. That also explains why Figure 9a and 9b coastlines are so similar.**

Author response #10:

Paleo-coastlines from Andren et al. 2011 were digitized and overlaid with our reconstructions. Since Andren's original maps do not contain exact coordinates, the digitization was done in a imprecise manner by roughly matching the geographic features.
We agree that the comparison is rather qualitative than quantitative due to georeferencing difficulty and mismatch in the data resolution between Andren et al 2011 and our study. Therefore we will remove this figure and replace by text descriptions. Further, we will also add comparison with other reconstructions such as Jakobson et al. (2007).

11. **There are some spelling errors in references and reference list and not all reference are on the list. In reference list sometimes only first author is shown are not.**

Author response #11:

We will double-check all the citations and reference list.

12. **In the following are some comments by line numbers.**

**Chapter 1. Introduction**

**45 climatically controlled eustatic sea level changes**

Author response #12:

Thank you. This will be corrected.

**46 Lambeck 2010 not in reference list**

Author response #13:

The reference will be added, thank you for spotting that.

**Chapter Geological setting some terminology is not clear/correct**

**62-63 Danish Straits and Swedish Sound?? First term is enough as it includes all straits,**

**second is The Sound or Öresund**

Author response #14:

Thank you for pointing this out. We will use „Danish Straits" to descibe all straits.

**64,68 Russian Plate - there is no such plate, do You mean East European (Russian) Plain?**

Author response #15:

Terminology will be corrected according to definitions given by Uscinowicz (2014) and Maystrenko et al. (2008).

**65 what are Western and Eastern European Platform?**

Author response #16:

We are aware of terminological discrepancies in structure description Northeast and Southwest of the SST/TTZ suture zone. We will change the terminology according to definitions given by Uscinowicz (2014) and Maystrenko et al. (2008).

**78, 85, 99, 143 the term inland ice should be replaced by glacier/icesheet**

Author response #17:

It will be replaced following your suggestion.

**82. sea-level drop What about if land uplift is smaller than sea level rise?**

Author response #18:

This sentence describes the interplay between two driving forcing: eustacy and isostasy. We used two examples to explain their relative importance in driving the connection/disconnection between the Baltic and the open North Sea. The case Reviewer #3 mentioned may lead to a connection between the basin and the open sea. We will rephrase these descriptions to increase their comprehensibility.

**83,84 not clear what is meant by gate and gate function its more like technical term used in artificial reservoirs not for natural waterbodies**

Author response #19:

We will rephrase the wording to avoid such confusion.

**85 That sentence is not clear and not correct as there are surely sediments and proxy data older than post-glacial period (=last 11700 cal yr BP)**

Author response #20:

There are indeed older data. However, due to several glacier advances much of this sediment was partly or completely eroded. Therefore, sediment and proxy data older than post-glacial period is

incomplete and much more scarce than the post-glacial period. We will rephrase this sentence for clarification.

**89 There is no LGM on Fig. 1**

Author response #21:

The description will be corrected to „The evolution of the Baltic Sea since the first stage of the Yoldia Sea at 11.7 kyr BP (Fig.1) is…"

**100 Heinsalu and Veski were using brackish-water Yoldia Sea**

Author response #22:

Thank you for correcting this. This will be corrected in the revised version.

**102 why so-called?**

Author response #23:

We will remove the word „so called".

**110. Figure 1 longitude values are wrong. Maps are difficult to read, because its not clear what is light blue and what is blue in Gulf of Botnia and near Oslo fjord. Baltic Ice Lake existed in Pleistocene**

Author response #24:

Thank you for pointing this out. We will improve the plots in a revised version accordingly.

**Chapter 3. Data and methods**

**144 the sentence meaning not clear**

Author response #25:

We agree that it is unclear. We will remove this sentence since it does not contain meaningful information related to our study.

**150 Figure 2 What ice model was used?**

Author response #26:

It was ICE-5G. We will add this information.

**Chapter 3.2. Eustatic data (EC)**

**166 Waelbroeck et al 2002 not in reference list**

Author response #27:

The missing reference will be added.

**168 global ocean? what in none global ocean?**

Author response #28:

You are right. We will replace the word „global" with „open".

**Chapter 3.3. Vertical crustal movements…**

**191-195 add here some references**

Author response #29:
References will be added.

**205 explain how You get 500 years timeslices for reconstructions if GIA resolution is 1000 years**

Author response #30:

It was achieved by assuming a linear trend between millenial-step reconstructions. It will be explained in the revised text.

**Chapter 3.4 Sediment thickness**
**That full section needs more explanation and some information about reliability and resolution of used data sources.**

Author response #31:

Information about resolution of input datasets as well as their associated uncertainty will be added as a new table in the revised version for clarification.

**Chapter Discussion**

**399 Rosentau et al 2021 black curves are not proxy-based but modelled by ICE-6G_C(VM5a)**

Author response #32:

Thank you for pointing this out. This will be corrected.

**435 Figure 9 Andren et al 2011 coastlines are not similar to published maps.**

Author response #33:

Pelease see our response #10.
We agree that the comparison is rather qualitative than quantitative due to georeferencing difficulty and mismatch in the data resolution between Andren et al 2011 and our study. Therefore we will remove this figure and replace by text descriptions. Further, we will also add comparison with other reconstructions such as Jakobson et al. (2007).

**439-440 that was already in chapter 4.2**

Author response #34:

Indeed. The purpose of repeating these numbers here is for comprehensibility of the narrative of the entire subchapter, so that readers do not need to go back to chapter 4.2 for the exact numbers.

---

## Author Response (AR1)

**egusphere-2024-1931**

**Documentation of changes and reply to the review comments**

*[The original review comments are in **bold and italic**]*

***REVIEWER COMMENTS:***

**Referee 1**

*This manuscript presents a marginal sea basin-scale reconstruction of palaeogeography based on the combination of a big dataset including the eustatic sea levels, the GIA and the Holocene sediment accumulation for the Baltic Basin. A comparison between the modelled results and proxy-data-based reconstruction can improve the GIA model parameters, which is important for the study on Holocene sea-level history. I only have minor comments as below:*

*1. Line 130, please explain why ΔSED should be included in the equation of relative sea level.*

Author response #1:

Sorry, it was our mistake in the writing process. ΔSED should not be included into eq. (2). This term was not included in our calculation of the relative sea level.

We have corrected the mistake in the revised version.

*2. line 305, constant sediment accumulation rate was assumed in the estimation of sedimentation thickness at each time slice. However, sedimentation rate varies largely because of the changes in relative sea level, sediment supply, etc. particularly in coastal zone. Authors made a discussion in the last section of 5.3 for effect of the sediment dynamics. I feel this is not enough and expect an evaluation of the spatial distribution of areas characterized by changing sedimentation rate.*

Author response #2:

We are aware of the fact that the sediment accumulation rates varied throughout the Holocene. Even though these rates could be estimated based on analysis of sediment cores (being point data), extrapolating a few tens of existing point data to the entire Baltic Sea basin would be difficult to justify as sedimentary environments of the Baltic vary not only in time but also in space. Moreover in the Baltic Sea, the magnitude of ΔSED remains relatively small compared to the magnitudes of ΔEC and ΔGIA. Also, the highest sedimentation rates are situated in the deeper basins. Therefore assuming different sedimentation rates would have only minor influence on the paleo-bathymetry. We have added discussion on this in the revised version based on the above-mentioned arguments in the revised section 5.3:

*„A future challenge towards improvement of paleogeographic reconstructions requires differentiation of sediment accumulation and erosion rates. Although sediment accumulation rate may vary throughout the Holocene, we adopted a constant rate in this study due to poor data constraint. The reported rates were mostly derived based on analysis of sparsely distributed sediment cores and it is difficult to extrapolate a few tens of point data to the entire Baltic Sea. Therefore, more measurement data is needed to provide a sound database for extrapolation. Mass-balanced reconstruction methods have been proposed for backstripping of depositional areas and backfilling of erosional areas at geological time scales (Hay et al., 1989; Feng et al., 2023). However, these are of high uncertainty for reconstructions at a millennial scale due to insufficient resolution in the seismo-acoustic profiles. An*

*attempt to reconstruct the eroded coastal landscape in mid Holocene has been done by Zhang et al. (2014) at a local scale. The reconstruction was based on an extrapolation of the shape of present-day remnants of erodible coast by a fitted spline function using the morphology of the backland as a reference. Based on the identified major source and sink terms as well as the associated transport pathways exemplified in this study, it might also be feasible to apply the same method to reconstruct eroded coasts at a centennial-to-millennial time scale. "*

It is worth to note that our attempt is one of the first complex applications integrating ΔSED to paleogeographic reconstructions at a marginal sea scale. As pointed out in discussion, applying it to more sedimentation-dominated environment, such as SE Asian shelf, would require slightly different approach by taking into account sediment compaction, erosion and different accumulation rates.

*3. line 401, "whereas the Vistula Spit curves shows continuous sea-level fall" should be "whereas the Vistula Spit curves show continuous sea-level rise".*

Author response #3:

This mispelling has been corrected. Thank you for pointing it out.

**Referee 2**
*General comments to the submitted manuscript: 'Paleogeographic numerical modelling of marginal seas for the Holocene – an exemplary study of the Baltic Sea' by Miluch et al.*

*The paper describes the paleogeographic distribution of the Baltic Sea during the Holocene. They base their observations on different data sets gathered from different countries along the Baltic Sea. For the reconstruction, they mainly use three parameters: vertical changes of the water level, vertical changes of the landmasses (tectonics and isostasy), and the thickness of the sediment deposits.*

*The novelty is that they include the latter one. The paper is well structured, and the Figures are certainly relevant for the message.*

*After reading the manuscript I have some general discussion points left:*

1. *The spatial resolution of the grid size in the Baltic is not always clear. I have no idea how accurate and variable for instance the input data of the isostacy is, nor of those of the sediment thickness data. In qualitative sense, they indicate for instance at the data in the northern Baltic Sea is sparse, but I have no idea how sparse (in a quantitative sense, how many data points per km²).*

Author response #4:

The grid size is uniform at 0.01° x 0.01°. All point data are interpolated to this grid. The initial digital elevation model derived from the latest product of GEBCO has the resolution of 15 arc seconds (0.004167° x 0.004167°). The GIA dataset was initially at a global-scale with 0.5° x 0.5° grid size. It was then extracted for northern Europe and interpolated to the 0.01° x 0.01° grid.

Sediment thickness data were interpolated based on several original datasests from difference sources and with different spatial resolution, including gridded data for local parts, seismic profiles and point measurements. These source data were used for trainning and validation for the machine learning approach (U-net) which was used to fill the data gaps (extrapolation) in other parts of the study area. In addition, co-krigging method was also applied for interpolation at some local parts

where existing measurements show high correlation between sediment type/water depth and sediment thickness.

We have added information about details of each sub-dataset as a new table (Table 1) in the revised version for clarification.

> 2. *The spatial interpolation and extrapolation techniques are not always explicitly mentioned. A Table for all parameters including it's resolution and spatial interpolation technique would be nice.*

Author response #5:

We agree. A table (Table 1) providing information of each sub-dataset as well as the interpolation and extrapolocation techniques has been added in the revised version for clarification.

> 3. *The discussion about the different phases is much based on the glacial history of the area. While reading the paper, I wondered how large the influx from major rivers in central Europe was, and how this differed during the Holocene. These rivers will certainly transport sediments towards the southern shores of the Baltic.*

Author response #6:
We have added a paragraph to discuss the role of influx from major rivers in the revised section 5.2:

*"Coastal erosion, namely erosion of the glacial-till cliffs, serves as the main source that contributes at least 80% of the annual deposition in the basins (Wallmann et al., 2022). Sediment supply from major central European rivers, namely Oder, Vistula, Nemunas, Daugava and Neva, is on the order of $1 \times 10^7$ t yr$^{-1}$ (Porz et al., 2021; Pruszak et al., 2005; Lajczak and Jansson, 1993) with the largest contribution from the river Vistula (0.16-0.4×10$^7$ t yr$^{-1}$). This indicates that the riverine sediment supply accounts for less than 10% of the total Holocene sediment budget in the Baltic Sea. Biogenic production contributes to 4-15% of the annual deposition budget (Wallmann et al., 2022; Porz et al., 2021). In addition, sediment input from the North Sea is estimated to be on the order of $1 \times 10^6$ t yr$^{-1}$, contributing to 10-30% of the annual deposition in the SW Baltic Sea (Porz et al., 2021). However, this input is almost negligible (accounting for ~1%) compared to the total annual accumulation rate in the Baltic Sea. Additional supply of sediment from melting of the ice cover on the Scandinavia may occur during early Holocene stages. However, such supply might also be negligible compared to the total Holocene sediment budget given that the Scandinavian mountains provide only a very small suspended sediment yield that is on the order of $1 \times 10^5$ t yr$^{-1}$ (Lajczak and Jansson, 1993). "*

> 4. **The interpretation around the Danish straits is based on very few data and probably also modeled in a very course sense. The authors discuss their results with those over earlier studies and conclude that not all studies for instance have two straits. How comparable are these studies if both studies are based on not a lot of observations, nor have a detailed modeling grid.**

Author response #7:

The timing of events such as opening of straits varies between reconstructions. Some local-scale variabilities of narrow structures such as straits may not be well visible on the regional maps due to coarse map resolution, difference in the data coverage and/or reconstruction methods. Therefore we agree that it is not comparable between regional studies on local-scale features which do not have sufficient data coverage. We have modified the discussion on this in the revised version (section 5.1):

*"We also compared our modelled paleo-DEMs with the maps from Andrén et al. (2011) in a qualitative manner and identified a general consistency between the two sets of maps in the location of the gates between the Baltic basin and the open North Sea as well as the timing of their closing*

*and opening (section 4.3)... ... Our reconstructed morphology of the Littorina Sea at 6.5 kyr BP is characterized by an open connection between the Baltic Sea and the North Sea through the Danish Straits, which persists until today. In the reconstruction by Andrén et al. (2011), the Baltic-North Sea connection at 6.5 kyr BP exists only via the Great Belt, whereas all three straits are already opened in our result at that time. The timing of events such as opening of straits varies between reconstructions. Some local-scale topographic structures such as straits may not be well resolved in regional reconstructions due to insufficient spatial resolution or data coverage. Similar to earlier scenarios, the RSL in the northern and central parts of the Baltic Sea is generally lower in the map of Andrén et al. (2011) compared to our result. „*

5. **The novelty is concentrated on the inclusion of sediment thickness data for the Baltic Sea. In the manuscript, the authors talk about 'sediment dynamics', 'sediment budget', 'sediment mass', and 'sediment thickness'. The paper becomes much more precise when a clear terminology is chosen. Really like your terms 'thickness' or 'mass'. Perhaps even better 'thickness' and 'vertical changes in thickness'. 'Budget' is perhaps more than you present (with a budget, you normally also mention lateral fluxes in the system). 'Dynamics' is closely related to sediment transport resulting in areas with erosion or deposition and you are not showing that.**

Author response #8:

We refer to different meanings when using different terminologies. 'Sediment dynamics' corresponds to dynamic processes including sediment transport, deposition and erosion. We used this term to highlight the novelty of our method by including the impact of sediment deposition on the paleo-morphology and explain why integrating sediment dynamics is important for paleogeographic research. 'Sediment mass' is an overall mass of sediments calculated from a grid, whereas 'sediment thickness' refers to the vertical thickness of sediment deposition for the period of interest. 'Sediment budget' refers to detailed analysis regarding sediment sources, sinks and transport pathways. Clarification of each term will be added in the revised manuscript. We agree that lateral fluxes such as riverine input, coastal erosion and sediment exchange between the North Sea and the Baltic Sea should be included in the sediment budget analysis, and we have added estimates of these fluxes in the revised version in section 5.2:

*"Coastal erosion, namely erosion of the glacial-till cliffs, serves as the main source that contributes at least 80% of the annual deposition in the basins (Wallmann et al., 2022). Sediment supply from major central European rivers, namely Oder, Vistula, Nemunas, Daugava and Neva, is on the order of $1\times10^7$ t yr$^{-1}$ (Porz et al., 2021; Pruszak et al., 2005; Lajczak and Jansson, 1993) with the largest contribution from the river Vistula (0.16-0.4$\times10^7$ t yr$^{-1}$). This indicates that the riverine sediment supply accounts for less than 10% of the total Holocene sediment budget in the Baltic Sea. Biogenic production contributes to 4-15% of the annual deposition budget (Wallmann et al., 2022; Porz et al., 2021). In addition, sediment input from the North Sea is estimated to be on the order of $1\times10^6$ t yr$^{-1}$, contributing to 10-30% of the annual deposition in the SW Baltic Sea (Porz et al., 2021). However, this input is almost negligible (accounting for ~1%) compared to the total annual accumulation rate in the Baltic Sea. Additional supply of sediment from melting of the ice cover on the Scandinavia may occur during early Holocene stages. However, such supply might also be negligible compared to the total Holocene sediment budget given that the Scandinavian mountains provide only a very small suspended sediment yield that is on the order of $1\times10^5$ t yr$^{-1}$ (Lajczak and Jansson, 1993). „*

6. **More specific comments are included in the pdf-file of the manuscript.**

Author response #9:

Thank you for the attached comments. We have implementated modifications to text based on the comments. Here a couple of answers to your important questions or comments:

Line 42. We decided to leave 'post glacial period' here. We believe it's clear enough and it's not strictly limited to very beginning of Holocene.

Line 56. Above we discuss factors in general. Here we discuss details of model components.

Lines 146 and 147. These sentence discusses in general tools of the software, which comprises of various interpolation methods and allows to generate grids with pre-defined resolution.

Line 172. It is not specified however we know that this region, due to being situated on the hinge-line is the least-GIA-influenced area of the Baltic.

Line 278. It's purely random. This method is characterized by little randomness.

Line 299. Sediment budget analysis, discussing the sediment sources is conducted in chapter 4.2.

Line 305. Although sediment accumulation rate may vary throughout the Holocene, we adopted a constant rate in this study due to poor data constraint. The reported rates were mostly derived based on analysis of sparsely distributed sediment cores and it is difficult to extrapolate a few tens of point data to the entire Baltic Sea. Therefore, more measurement data is needed to provide a sound database for extrapolation. Moreover at the Baltic Sea regional scale, ΔSED remains as a minor component of the equation in comparison to ΔEC and ΔGIA. Also, the highest sedimentation rates are situated in the deeper basins. Therefore assuming different sedimentation rates would have very minor influence on the paleo-bathymetry.

Our attempt is one of the first applications of this method. As pointed out in discussion chapter, applying it to more sedimentation-dominated environments, such as the SE Asian shelf, would require a slightly different approach by distinguishing the impact of sediment erosion and accumulation rate in different periods. However, on the other hand, incorporating complicated paleo-morphodynamic models of sediment accumulation, erosion, redeposition or compaction as well as sediment fluxes will bear high uncertainty due to insufficient data for model forcing, boundary configuration, calibration and validation. This needs to be addressed in future studies.

Line 369. These locations mark the connections between the Baltic and the North Sea. We refer to them in the text, therefore we decided to place them on the map.

Line 436. Indeed. Paleo-cliffs may be found inland of Wolin Island (Poland) nearby Szczecin Lagoon shoreline (it was a marine bay during Littorina trangression). Also terraces of paleo-coastlines are well-seen on the Gotland Island, however, in this case it is mainly related to isostatic uplift.

Line 457. In this paragaph we compare present sediment accumulation rate in the SW Baltic Sea from Porz et al. (2021) with the number referred from our modelled sediment thickness in this region. Major sources and sinks are further described in the following paragraph in our revised version.

Line 463. We would rather leave the word „dynamics" here. Even though for the Baltic Sea study we simply subtract the thickness grid, the following chapter acts as an outlook for future studies. We discuss application of this method to more sedimentation-dominated environments in which sediment dynamic processes (erosion, accumulation, redeposition, sediment fluxes) need to be taken into account, or application of it in a local scale.

Line 467. There are spits like this in the Baltic coast: Hel Peninsula (Poland), Vistula Spit (Poland/Russia) or Curonian Spit (Russia/Lithuania). We would rather keep the sentence and add examples into the text.

Line 509. In this sentence we summarize what is a new achievement of our study and discuss application of consistent methodology of paleogeographic modeling into different climatic zones. There are no new material here.

**Referee 3**

1. **General comments to the manuscript.**
   **The manuscript deals with paleogeography of the Baltic Sea during the Holocene by combining eustatic sea-level change, glacio-isostatic adjustment and sediment deposition.**

   **They present paleoreconstruction and map of Holocene sediment thickness based on different datasets and calculate total mass of Holocene sediment in the Baltic Sea and yearly sediment accumulation.**

   **Reading the manuscript there are terminological and methodological problems, which are described below.**

   **First there are some errors using terminology, like using plate (plate tectonics) instead plain and its not clear what is meant by platform (lines 64-68 see comments below). For some terms it's not clear the meaning, like inland ice (possibly glacier), gate and gate function or amphibious Digital Elevation Model.**

Author response #10:

Thank you for the comment. We are aware of terminological discrepancies in structure description Northeast and Southwest of the SST/TTZ suture zone. We have modified the terminology according to definitions given by Uscinowicz (2014) and Maystrenko et al. (2008) in the revised version (first paragraph of section 2).

We deliberately chose the term "gate" to describe the hydrographic interaction of sea level rise and vertical coastal movement in opening and closing the connecting routes between the Baltic Sea basin and the Paleo-North Sea. This term is widely used in literature for such topographic features. We have added an explanation in the revised version (line 85-86, second paragraph of section 2).

„Inland ice" stands here indeed for glacier. We have changed the term accordingly. „Amphibiuous Digital Elevation Model" refers to a DEM that covers both subaquous and subaerial parts. We have added an explanation and removed the word „amphibiuous" to avoid confusion.

2. **Secondly there are some methodological problems like Baltic Ice Lake /Yoldia Sea transition (look details below Figure3) and creation of sediment thickness map.**
   **Chapter 3.4 Sediment thickness does not have information about the uncertainties of the used data sources. Why only present-day sea area data were used? In central and northern areas, like Gulf of Botnia, coastline was several hundred meters higher, and sedimentation in the Baltic Sea occurred also in present day mainland. Moreover referred Winterhalter 1972 does not have any datapoints from north of Gulf of Botnia (yellow square in Figure 5) so its not clear how those data were manipulated.**

Author response #11:

Information about uncertainties and input data resolution has added as Table 1 in the revised manuscript (see also Author response #4). Regarding sedimentation on present-day mainland (mainly the northern Baltic coast) after it was emerged, this part of sedimentation has been subjected to reworking by both water flow and wind, therefore any estimation of marine sediment thickness would be even more difficult to obtain and justify. Moreover, the Holocene sediment thickness of the northern Baltic Sea and its coast is generally very thin that any extrapolation to the mainland would have little impact on the paleo-DEM and negligible impact on the total estimated sediment budget of the Holocene. We have included discussion of this aspect in the revised manuscript (last paragraph, section 5.2):

*„It is worth to note that our Holocene sediment thickness map is spatially confined by the present-day Baltic Sea coast. Although Holocene sedimentation may also occur on parts of the present-day mainland especially in the northern Baltic coast when they were submerged, the deposited sediment was subjected to reworking when these parts became emerged and therefore is difficult to quantify due to lack of data. Moreover, the Holocene sediment thickness of the northern Baltic Sea and its coast is generally scarce (Fig.6), therefore omission of deposit on the mainland is considered to have minor impact on the paleo-DEMs and the total Holocene sediment budget in the Baltic Sea."*

Regarding Winterhalter 1972 dataset it was indeed limited to the southern Gulf of Bothnia. Unfortunately we could not find any literature providing sedimentation thickness in the northern Gulf of Bothnia. This is why we used co-kriging to extrapolate it to the north. A brief explanation has been added in section 3.4.4 for clarification.

3.  **According to line 231, Holocene sediments in the southern Baltic Sea are on top of the glacial till. Holocene started at ~11.7 ka BP but glacial till accumulated around 17-15 ka BP, so there was no sedimentation several thousand of years? According to line 241 glacial varved sediments of the Baltic Ice Lake are considered early Holocene age, what is not true as Baltic Ice Lake drainage (end) coincides more or less with the start of Holocene, so Baltic Ice Lake sediments are from Pleistocene, not from Holocene.**

Author response #12:

Top of glacial till is a well-visible seismic reflector, whereas the boundary between the Baltic Ice Lake (BIL) and Yoldia Sea is hard to determine from the seismic profiles. Due to the fact that BIL sediment chronostratigraphically belongs to Late Pleistocene, the modelled thickness may be slightly overestimated. We have added an explanation to section 3.4.2 including an estimate of uncertainty related to this:
*„...The top of glacial till is a well-visible seismic reflector on most profiles, whereas the boundary between the BIL and the Yoldia Sea is difficult to discern in some profiles. Since the BIL belongs to the late Pleistocene, the thickness for the Holocene sediment may be slightly overestimated in these profiles. Maximum local thickness of BIL sediments up to 7 m is found in the deepest parts of the basins and diminishes towards the shallower areas according to Christiansen et al. (2002). Therefore, our estimated thickness of the Holocene sediment might be overestimated by ~20% in the deepest part of the basins, with the uncertainty decreasing towards the shallower areas...."*

In addtion, we have also estimated uncertainty in the sediment budget related to this in section 4.2 of the revised version:
*"...As described in section 3.4.2, unclear boundary between late Pleistocene and early Holocene sediment in the seismic profiles across the deep basins (e.g. the Gotland basin) may lead to*

*overestimation of the Holocene sediment thickness by ~20% in the deepest part of the basins. The overestimation decreases toward shallower areas. A uniform overestimation of the deposition thickness in the basins by 20% corresponds to a sediment budget of $1.1 \times 10^{11}$ t, which is ~8% of the estimated mean total budget ($1.34 \times 10^{12}$ t)."*

4.  **There has been earlier attempt to create Holocene sediment thickness by Jakobson et al 2007 https://doi.org/10.1016/j.gloplacha.2007.01.006, which differs from results here. So the map presented here seems to include not only Holocene sediments but some Pleistocene sediments also.**

Author response #13:

According to the information from Jakobson et al. (2007), the Holocene sediment thickness map for the Baltic Basins shown in their Figure 3d was compiled through assembling information from available sediment distribution maps and information retrieved from the Swedish Geological Survey's mapping archives which unfortunately do not provide an open access. A comparison between our map (Figure 5) and the map from Jakobson et al (2007) shows a general agreement in the Borholm basin and along the Swedish coast near the Gotland. However, there exists a large descrepency in the thickness value in other basins (e.g. Arkona basin, Gotland basin) between the two maps. The thickness values in Jakobson et al. (2007) for these basins are much smaller than previous published values from Lemke (1998) and Uścinowicz (1998) focusing on these local areas. The compiled thickness data from the difference sources we have collected show more consistent patterns covering both deep basins and shallow coastal areas and therefore we argue that our map provides a more accurate distribution of Holocene sediment thickness compared to earlier publications. We have added discussion on this aspect in the revised version (last paragraph, section 5.1):

*„An earlier effort in mapping Holocene sediment thickness in the Baltic Sea has been made by Jakobson et al. (2007). The mapping was through assembling information from available sediment distribution maps and information retrieved from the Swedish Geological Survey's mapping archives which unfortunately do not provide an open access. The resultant map is characterized by relatively low spatial resolution and limited to the southern and central parts (without Bothnian Bay) of the Baltic Sea (Jakobson et al., 2007). A comparison between our map (Figure 5) and that from Jakobson et al. (2007) shows a general agreement in the Borholm basin and along the Swedish coast near the Gotland. However, there exists a large discrepancy in the thickness value in other basins (e.g. Arkona basin, Gotland basin) between the two maps. The thickness values in Jakobson et al. (2007) for these basins are much smaller than previous published values from Lemke (1998) and Uścinowicz (1998) focusing on these local areas. Despite a likely overestimation of the Holocene sediment thickness in the deep Gotland basin as pointed out in section 3.4.2, integrated data from the difference sources within this study show more consistent patterns covering both deep basins and shallow coastal areas and therefore provide a more accurate distribution of Holocene sediment thickness."*

5.  **There are some issues with Figures:**

**Figure 1 longitude values starting from 15° and specially 20°-40° are almost 5° wrong. Glacier extent specially for 10.5 ka BP is not the same as in Andren et al 2011. It seems that there is problem with georeferencing.**

Author response #14:

Thanks for pointing this out. We admit that it is a mistake in our georeferencing. An updated Figure 1 by adopting the same layout as the original maps by Andren et al. (2021) has been provided.

6. **Figure 2 according to figure Peltier 1999 ice thickness model was used (ICE-4G), but in text Peltier 2004 (ICE-5G)**

Author response #15:

Thanks for pointing out this mistake. It has been corrected to Peltier (2004) in Figure 2 in the revised version, which is now consistent with the descriptions in section 3.3.

7. **Figure 3 explains that authors have wrongly modelled Baltic Ice Lake at 11.7 ka BP or they don'tunderstand how Yoldia Sea Stage started. According to that figure the highest BIL water level occurred 12 ka cal BP and not at 11.7 ka BP as suggested by Andren et al 2011. At 11.7 ka BP water level in BIL dropped during ca 1-2 years 25 meters and Yoldia Sea started. So modelled BIL at 11.7 ka BP (Fig 7) is actually Yoldia Sea first stage after BIL drainage not BIL prior the drainage. In the 11.7 kyr BP map by Andren (2011), the hydrographic connection between the Baltic Sea basin and the Kattegat indicates communicating systems. This makes Andren's caption "(...) BIL prior to final drainage" questionable.**

Author response #16:

Please note that the red line on Fig 3. stands for the Uścinowicz (2006) RSL curve that we used only for the Ancylus lake phase. Our reconstruction started exactly at 11.7 kyr BP when the water level of the Baltic region was dropped to the level of the open North Sea. You are right that modelled BIL at 11.7 kyr BP is actually Yoldia Sea first stage after BIL drainage but not BIL prior the drainage. We have corrected „BIL prior to drainage" to „Baltic Ice Lake/Yoldia Sea transition" in all relevant texts including captions of Figure 1 and 7.

8. **Figure 7 reconstruction for 11.7 kyr BP and 11.0 kyr BP look in the southern part near Bornholm exactly same**

Author response #17:

They are similar but not the same. The RSL on the map 7b is higher than that on map 7a what makes the land bridge connecting Bornholm with mainland narrower.

**Figure 8 the caption is not correct. Both curves red (results in manuscript )and black (Rosentau et 2021) are modelled RSL curves according to ICE-5G model. Rosentau et al 2021 has on Figures 7,8,9 shown results with ICE-5G model with three different lithosphere thicknesses and also ICE-6G, which one is used here is not clear. Black curve is not field data (or proxy data). Only RSL for Finland N looks similar to Rosentau et al 2021 results.**

Author response #18:

Thank you for pointing it out. We have corrected Fig. 8 caption and removed the word „field data".

Rosentau et al. (2021) presented different curves of RSL using different model parameter setting (lithosphere thickness of 80, 100 and 100 km) and versions (ICE-5G, ICE-6G). We did not intend to compare our results with all curves from Rosentau et al. (2021). This is beyond the scope of this study. We digitized their curves based on the ICE-5G model with 120km lithosphere thickness, because these curves are close to the mean value of the range defined by all curves. This information is now added to the caption. We agree that the RSL of Rosentau et al (2021) show differences with our results mainly for the initial stage between 11.7- 10 kyr BP and afterward a general agreement is reached. We have added some descriptions and discussion on this aspect in the revised version (first paragraph, section 5.1):

*„...A comparison of RSL curves between those by Rosentau et al. (2021) and our results shows a general agreement except for the early Holocene period, as described in the previous section. In three stations, namely Finland N, Blekinge and Vistula Spit, our results show a lower RSL at 11.7 kyr BP than those in Rosentau et al. (2021) adopting 120 km lithosphere thickness (Fig. 8). It should be noted that there exist remarkable differences in the reconstructed local RSLs for the early Holocene period among the scenarios adopting different lithosphere thickness values as shown in Rosentau et al (2021). As pointed out by Rosentau et al. (2021), the reconstructed curves using global ICE-5G and ICE-6G_C ice histories overestimate the RSL and fail to capture a mid-Holocene high-stand (~7.5-6.5 kyr BP) inferred from the proxy data in the transitional area. This overestimation seems to originate from an overestimation of ice loading in the ICE-5G and especially in the ICE-6G_C models. Our modelled curves lie in the lower limit of the RSLs in Rosentau et al. (2021), and therefore may provide results closer to proxy data."*

9. **Figure 9 Comparison in present form is not convincing as shorelines from Andren et al 2011 seems to like freehand drawings and differ from original. Moreover, on Figure9 a) You compare results here with BIL prior final drainage (Andren et al 2011) but its water level was about 25 meter higher than in present reconstruction. That also explains why Figure 9a and 9b coastlines are so similar.**

Author response #19:

Paleo-coastlines from Andren et al. 2011 were digitized and overlaid with our reconstructions. Since Andren's original maps do not contain exact coordinates, the digitization was done in a imprecise manner by roughly matching the geographic features.
We agree that the comparison is rather qualitative than quantitative due to georeferencing difficulty and mismatch in the data resolution between Andren et al 2011 and our study. Therefore we have removed this figure and replaced by text descriptions. Further, we have also added a comparison with the reconstruction by Jakobson et al. (2007) in section 5.1 (see also Author response #13).

10. **There are some spelling errors in references and reference list and not all reference are on the list. In reference list sometimes only first author is shown are not.**

Author response #20:

We have double-checked all the citations and the reference list.

11. **In the following are some comments by line numbers.**

**Chapter 1. Introduction**

**45 climatically controlled eustatic sea level changes**

Author response #21:

Thank you. This has been corrected.

**46 Lambeck 2010 not in reference list**

Author response #22:

The reference has been corrected, thank you.

**Chapter Geological setting some terminology is not clear/correct**

**62-63 Danish Straits and Swedish Sound?? First term is enough as it includes all straits, second is The Sound or Öresund**

Author response #23:

Thank you for pointing this out. „Danish Straits" is now used to descibe all straits in the revised version.

**64,68 Russian Plate - there is no such plate, do You mean East European (Russian) Plain?**

Author response #24:

Terminology has been corrected according to definitions given by Uscinowicz (2014) and Maystrenko et al. (2008).

**65 what are Western and Eastern European Platform?**

Author response #25:

We are aware of terminological discrepancies in structure description Northeast and Southwest of the SST/TTZ suture zone. We have corrected the terminology according to definitions given by Uscinowicz (2014) and Maystrenko et al. (2008).

**78, 85, 99, 143 the term inland ice should be replaced by glacier/icesheet**

Author response #26:

It has been corrected following your suggestion.

**82. sea-level drop What about if land uplift is smaller than sea level rise?**

Author response #27:

This sentence describes the interplay between two driving forcing: eustacy and isostasy. We used two examples to explain their relative importance in driving the connection/disconnection between the Baltic and the open North Sea. The case Reviewer #3 mentioned may lead to a (re)connection between the basin and the open sea. We have rephrased these descriptions to increase their comprehensibility.

**83,84 not clear what is meant by gate and gate function its more like technical term used in artificial reservoirs not for natural waterbodies**

Author response #28:

We have rephrased the decription of gate in the revised version (line 84-86):

*„The relationship between the vertical crustal movement and the sea-level change determines the hydrographic communication between the open North Sea and the Baltic Basin. The parts connecting*

*the Baltic Basin and the North Sea thus serve a function of "gate" that is opened or closed by eustacy-isostacy-ice interaction."*

**85 That sentence is not clear and not correct as there are surely sediments and proxy data older than post-glacial period (=last 11700 cal yr BP)**

Author response #29:

There are indeed older data. However, due to several glacier advances much of this sediment was partly or completely eroded. Therefore, sediment and proxy data older than post-glacial period is incomplete and much more scarce than the post-glacial period. We have rephrased this sentence for clarification:

*„However, because of the erosional effects of the advancing ice sheet, sediments reflecting the geological history by proxy-data remained scarcely from the pre-glacial period."*

**89 There is no LGM on Fig. 1**

Author response #30:

The description has been corrected to:

*„Andrén et al. (2011) have depicted this postglacial history based on interpretation of proxy data including basin sediments and markers of paleo-coastlines by a set of paleogeographic maps (Fig. 1) which are used in our study for qualitative assessment of the results achieved by numerical modeling."*

**100 Heinsalu and Veski were using brackish-water Yoldia Sea**

Author response #31:

Thank you. This has been corrected in the revised version.

**102 why so-called?**

Author response #32:

We have removed the word „so called".

**110. Figure 1 longitude values are wrong. Maps are difficult to read, because its not clear what is light blue and what is blue in Gulf of Botnia and near Oslo fjord. Baltic Ice Lake existed in Pleistocene**

Author response #32:

Thank you for pointing this out. Figure 1 has been updated and information for interpreting the colors has been added.

**Chapter 3. Data and methods**

**144 the sentence meaning not clear**

Author response #33:

We agree that it is unclear. We have removed this sentence since it does not contain meaningful information related to our study.

**150 Figure 2 What ice model was used?**

Author response #35:

It was ICE-5G. This information has been added in the revised version.

**Chapter 3.2. Eustatic data (EC)**

**166 Waelbroeck et al 2002 not in reference list**

Author response #36:

The missing reference has been added.

**168 global ocean? what in none global ocean?**

Author response #37:

You are right. We have replaced the word „global" with „open".

**Chapter 3.3. Vertical crustal movements…**

**191-195 add here some references**

Author response #38:
References have been added.

**205 explain how You get 500 years timeslices for reconstructions if GIA resolution is 1000 years**

Author response #39:

It was achieved by assuming a linear trend between millenial-step reconstructions. It is now explained in the revised text.
*„…A time interval of 500 years required for this study was achieved by assuming a linear trend between each millennium."*

**Chapter 3.4 Sediment thickness**
**That full section needs more explanation and some information about reliability and resolution of used data sources.**

Author response #40:

Information about resolution of input datasets as well as their associated uncertainty has been added as Table 1. Please see also Author response#4

**Chapter Discussion**

**399 Rosentau et al 2021 black curves are not proxy-based but modelled by ICE-6G_C(VM5a)**

Author response #41:
Thank you for pointing this out. This is corrected in the revised version in caption of Figure 8. We would like to point out that Rosentau et al (2021) applied four GIA scenarios using global ICE-5G (with three different lithosphere thicknesses, namely 80 km, 100 km and 120 km) and ICE-6G_C (VM5a), and we compared our results with their result using ICE-5G with lithosphere thickness of 120 km. A discussion on the comparison is added in the first paragraph of the Discussion section. For details please see Author response #18.

**435 Figure 9 Andren et al 2011 coastlines are not similar to published maps.**

Author response #42:

Pelease see our response #19.
We agree that the comparison is rather qualitative than quantitative due to georeferencing difficulty and mismatch in the data resolution between Andren et al 2011 and our study. Therefore we have removed this figure and replaced by text descriptions. Further, we have also added a comparison with the reconstruction by Jakobson et al. (2007).

**439-440 that was already in chapter 4.2**

Author response #43:

Indeed. The purpose of repeating these numbers here is for comprehensibility of the narrative of the entire subchapter, so that readers do not need to go back to chapter 4.2 for the exact numbers. Please also note that we have restructured the sections and placed the uncertainty analysis of the sediment budget in section 4.2.

---

## Author Response (AR2)

**egusphere-2024-1931**

**Documentation of changes and reply to the review comments**

*[The original review comments are in **bold and italic**]*

***REVIEWER COMMENTS:***

**Referee 1**

*The manuscript is greatly improved after revision. However, there are three comments, which still need your attention.*

*1.      You extended Methodology describing available sediment thickness data, but its not completely clear how realistic are interpolations for areas where there are no data*

Author response #1:

We applied two types of extrapolation and interpolation for areas where there are no data, namely co-kriging and U-net (machine learning). Co-kriging was applied to areas where strong correlation between the target variable (Holocene sediment thickness) and the predictor variable (paleo-bathymetry) was identified, whilst U-net was applied to areas where there is no strong dependence of the target variable on any individual variables.

In section 3.4.4. the following information about correlation between the two variables used for co-kriging and corresponding reliability has been added (lines 268-269 in the revised version):

„The Pearson's correlation coefficient between the two variables is 0.52 in the measured point data, suggesting the feasibility of co-kriging-based extrapolation."

Subsection 3.4.5 provides the information on validation of the U-net extrapolation. The average deviation from the validation of the measured data is 5.8%, suggesting a highly satisfactory performance of the method (lines 282-289):

„In the 420 sub-datasets, 80% were used for model training and 20% for assessment of the model prediction. The input of the U-Net has the shape (32, 32, 4) and the output shape is (32, 32). The first layer consists of a double convolutional block performing 3×3 convolution with 64 output channels, padding, batch normalization and ReLU activation. The training was performed with 100 epochs and the mean squared error (MSE) was calculated as the loss function (torch.nn.MSEloss). Re-running the model with different random initializations and dropout yields different model results with the same general pattern but some local differences in sediment thickness. The result with the smallest value of MSE (6.1 m²) was chosen (Fig.6). This corresponds to an average deviation from the validation to the measured data of 5.8%."

*2. Discussion lines 449-455*

*Your palaeoreconstruction for 11700 cal yr BP is in fact beginning of the Yoldia Sea, but Andreen et al 11700 stage is last stage of the BIL. So they are not comparable as water levels differ 25 meters.*

Author response #2:

The reason for such significant difference between these two reconstructions has been explained in lines 454-457 of the same paragraph. We have added 25 m for a quantitative impression:

*„Such difference may be attributed to that the map of Andrén et al. (2011) represents still the late stage of BIL when the water level was higher by ~25 m in the Baltic basin than the North Sea, whereas our result corresponds to the beginning of the post-drainage phase when the water level of these two seas converged."*

**3. Discussion lines 461-463**

**statement that according to Andren et al2011 at 6500 cal yr BP there was connection only via Great Belt is not correct. If You look Andren et al 2011 Fig.4.8 and read page 89 last paragraph, then there is clearly said that Õresund Strait was the main mechanism behind the onset of the Litorina Sea.**

Author response #3:

You are right. We have now corrected the statement to (lines 465-468 in the revised version):

*„In the reconstruction by Andrén et al. (2011), the Baltic-North Sea connection at 6.5 kyr BP exists via all three straits, being consistent with our result. Some local-scale topographic structures such as the shape of the straits may vary between different reconstructions due to insufficient spatial resolution or data coverage."*